# A Catalogue of Structural Variation across Ancestrally Diverse Asian Genomes

Joanna Hui Juan Tan [1,49], Zhihui Li [1,49], Mar Gonzalez Porta[1,46,49], Ramesh Rajaby [1,47,49], Weng Khong Lim [1,2,3,4], Ye An Tan[5], Rodrigo Toro Jimenez[1], Renyi Teo[1], Maxime Hebrard [1], Jack Ling Ow[1], Shimin Ang [1], Justin Jeyakani[1], Yap Seng Chong[6,7], Tock Han Lim[8], Liuh Ling Goh [9], Yih Chung Tham[10,11], Khai Pang Leong[9], Calvin Woon Loong Chin[12,13], SG10K_Health Consortium*, Sonia Davila[4,14,15,48], Neerja Karnani[16,17,18], Ching-Yu Cheng[10,11], John Chambers[19,20,21], E. Shyong Tai [2,5,21,22], Jianjun Liu [23,24], Xueling Sim [5], Wing Kin Sung[1,25,26], Shyam Prabhakar [1,27,28] ✉, Patrick Tan [1,2,3,21] ✉ & Nicolas Bertin [1] ✉

Structural variants (SVs) are significant contributors to inter-individual genetic variation associated with traits and diseases. Current SV studies using whole-genome sequencing (WGS) have a largely Eurocentric composition, with little known about SV diversity in other ancestries, particularly from Asia. Here, we present a WGS catalogue of 73,035 SVs from 8392 Singaporeans of East Asian, Southeast Asian and South Asian ancestries, of which ~65% (47,770 SVs) are novel. We show that Asian populations can be stratified by their global SV patterns and identified 42,239 novel SVs that are specific to Asian populations. 52% of these novel SVs are restricted to one of the three major ancestry groups studied (Indian, Chinese or Malay). We uncovered SVs affecting major clinically actionable loci. Lastly, by identifying SVs in linkage disequilibrium with single-nucleotide variants, we demonstrate the utility of our SV catalogue in the fine-mapping of Asian GWAS variants and identification of potential causative variants. These results augment our knowledge of structural variation across human populations, thereby reducing current ancestry biases in global references of genetic variation afflicting equity, diversity and inclusion in genetic research.

Human genomic variation plays a critical role in health and disease, making its study a vital area of biological and medical research[1,2]. To improve our understanding of genetic variation across diverse human genomes and populations, international consortia such as the 1000 Genomes Project[3] (1000G), Genome Aggregation Database (gnomAD)[4], and national efforts such as the U.K. 100,000 Genomes Project[5] and NIH's All of Us program[6] have reported large-scale population-based sequencing efforts to comprehensively delineate common and rare genetic mutations across different geographies and ancestry groups. Most of these studies have focused primarily on base-pair level variations such as single-nucleotide polymorphisms (SNPs) and short insertions/deletions (indels)[3,4,7]. Recently, structural variants (SVs) have emerged as another important source of variation[8,9]. SVs are genome rearrangements ≥50 bp and can be classified into different classes such as deletions, duplications, insertions (including mobile element insertions (MEIs)), translocations and inversions[10]. Different classes of SVs have been proposed to arise through various mechanisms, including non-allelic homologous recombination or MEI events[11].

A full list of affiliations appears at the end of the paper. *A list of authors and their affiliations appears at the end of the paper.

✉e-mail: prabhakars@gis.a-star.edu.sg; tanbop@gis.a-star.edu.sg; Nicolas_Bertin@gis.a-star.edu.sg

With the availability of whole-genome sequencing (WGS) and the development of SV calling algorithms, researchers are increasingly leveraging short-read WGS data to characterise the spectra of human SVs. In 2015, the 1000 Genome Project [12] analysed 2504 low-pass genomes (~7x coverage) to discover 68,818 SVs affecting 2.5x more base pairs in the genome compared to SNPs. The gnomAD-SV project[10] identified 335,470 SVs from 14,891 WGS samples, clarifying the impact of SVs in different portions of the genome and generating SV catalogues to facilitate the identification of SVs associated with medical and phenotypic traits. Some phenotypically/medically relevant SVs include Chr17p11.2 duplications leading to *PMP22* gene overexpression and Charcot-Marie-Tooth disease (an inherited neurological disorder)[13], and Chr7 deletions affecting the *ELN* (Elastin) gene associated with Williams neurodevelopment syndrome [14]. Some SVs may be pleiotropic, such as the aforementioned Chr7 deletions which are associated with autism[15], schizophrenia[16] and cancer[17]. Knowledge of SVs can also improve our understanding of human evolution, as some SVs display population and ancestry-specific patterns[10,12]. For instance, amylase, a key enzyme involved in the digestion of starch has a higher copy number in Asian populations where rice (starch) is a staple food[18]. These studies highlight the importance of characterising the diversity of SV landscapes on a global scale.

Asia accounts for 60% of the world population. However, many large-scale SV profiling projects have focused on individuals of European ancestry, resulting in an under-representation of SVs reflective of Asian populations (gnomAD-SV: 1304 Asian genomes; 1000 Genomes Project: 993 Asian genomes). Moreover, despite recent efforts to close this gap, current SV studies of Asian populations are still of limited sample size and have focused on single ancestry groups[19,20].

Singapore is a multi-ancestry country populated by individuals of Indian, Chinese and Malay ethnicity due to its immigration history. The majority of the residents (~74%)[21] in Singapore are Chinese, who are mainly descendants of Han Chinese from the southern provinces of China[22]. Malays represent 13.6%[21] of the population forms the second largest ethnic group in Singapore. The Malay community in Singapore are mainly descendants of Austronesian people in Southeast Asia, particularly from Malaysia and Indonesia. Lastly, Indians form the third largest ethnic group in Singapore. Majority of the Indians in Singapore are descendants of Indian migrants from south-eastern part of India[22]. Given the genetic diversity of the population, Singapore can serve in the first approximation as a snapshot of East Asian, South-East Asia, and South Asia populations, and is uniquely suited for cataloguing Asian SV landscapes and genomic variation.

The Singapore Genome Variation Project (SGVP)[22], the SG10K_Health[23] and the SG10K_Med[24] projects, which focussed on small variants (SNP and lesser than 50 bp long indels) have previously demonstrated the value of Singaporean genomes for precision medicine. Here, we describe one of the first and to our knowledge the largest multi-ancestry study of SVs in Asians. Using WGS data from 8392 individuals (SG10K_Health) along with specialised SV-calling tools, we identified and characterised SVs in these three Asian populations and related these SVs to regulatory and biological effects. Our results contribute to the growing body of research on SVs and fill a critical gap in deciphering the genomic variation landscape across Asian populations.

## Results

### SV catalogues of three major ancestry groups in Singapore

We analysed Illumina short-read WGS data of 9770 samples from the SG10K_Health study[23], comprising participants of Chinese (58%), Indians (24%) and Malays (18%) ethnicities. After CRAM-level quality control (QC) and removing samples failing at least 1 of 9 QC metrics (Methods), 8392 samples were retained. This data set is subsequently referred to as SG10K Structural Variant release 1.4 ("SG10K-SV-r1.4").

The SG10K-SV-r1.4 dataset comprises multiple sub-cohorts sequenced at heterogeneous depths and using different library construction methods (Supplementary Data 1). Previous studies have demonstrated that library preparation methods, PCR-free (PCR-) and PCR-amplified (PCR+), can cause non-uniformity of sequencing coverage[10], which can in turn affect the ability to accurately detect structural variation. Differences in sequencing depth between libraries within a collection also impact structural variation genotyping sensitivity. To ensure robust SV analysis and to reduce technical confounding factors, we split the collection into three datasets, namely (1) Discovery cohort of 5487 individuals (average sequencing depth: 15x, library construction method: PCR+), (2) 15x_validation cohort containing 1523 individuals (average sequencing depth: 15x, library construction method: PCR-), (3) 30x_validation cohort consist of 1922 individuals (average sequencing depth: 30x, library construction method: PCR+). We focused on the discovery cohort which contains the largest number of individuals with a uniform sequencing depth and library construction method to identify SVs. We used the two validation datasets to re-genotype the variants detected in the discovery so as to ensure that results observed in the discovery dataset are reproducible. Overall, when confined to the discovery cohort alone (n = 5487), this study represents one of the largest Asian SV studies to date (Fig. 1a), covering 4 times as many individuals of Asian ancestries compared to previous studies[10,12]. In addition, our discovery cohort contains individuals of Southeast Asian ancestry (1144 individuals of Malay ethnicity), a population which has to date not been included in previous large population-based SV studies[10,12].

For the SG10K-SV-r1.4 discovery callset, we focused on the three most common SV types: deletions, insertions, and duplications (Fig. 1b and Supplementary Fig. 1, "Methods" section). Due to their distinct genomic properties, it is challenging to accurately identify SVs using a single analytic tool[25], and most previous SV cataloguing efforts have employed a combined suite of SV class-specialised algorithms[10,12]. At present, there are a plethora of SV detection tools available, each with its own pros and cons. In order to identify the tools to generate the SG10K-SV catalogue, we benchmarked several well-known SV callers, including Manta[26], Delly[27] and Smoove[28]. SVs identified using long-read WGS in 34 1000G samples by Ebert et al.[29] were used as a truth set to assert the performance of each SV caller to recover joint-genotyped SVs across matched 30x and 15x down-sampled short-read WGS (Supplementary Note 1, Supplementary Data 2). While measures of precision for Delly were superior to that obtained with Manta, Manta yielded overall higher F1-scores than other tools individually or in combination (Fig. 1c–e and Supplementary Figs. 2 and 3). This benchmarking also allowed us to estimate the fraction of SVs missed by our SV detection pipeline between 15x and 30x WGS. On average, across all the 1000G samples, 14.6% of long-read-defined SVs re-identified when sequenced at a depth of 30x could not be re-identified when down-sampled to 15x (Supplementary Fig. 3). Although Manta performs better than other tools for deletions and insertions detection, it has inherent limitations to accurately detect duplications in regions containing tandem repeat sequences (e.g., microsatellites and minisatellites)[30,31]. We thus complemented Manta with SurVIndel2[32], an in-house developed algorithm that has demonstrated the ability to detect duplications at high sensitivity in such context (Supplementary Note 2, Supplementary Figs. 4 and 5, and Supplementary Data 3 and 4). Similarly, mobile elements insertions (MEIs) present characteristics for which the MELT[33] algorithm has demonstrated superior detection ability. Combining all three tools, we identified 73,035 SVs comprising 29,011 insertions (including MEIs), 11,560 deletions, and 32,464 duplications. Approximately 66.5% and 86.7% of SG10K-SV-r1.4 events were novel (Fig. 2a, b) with respect to gnomAD-SV[10] and 1000 Genomes Project phase 3 SV[12] (1000G-SV), respectively, reflecting the potential for new discoveries when analysing underrepresented Asian genomes.

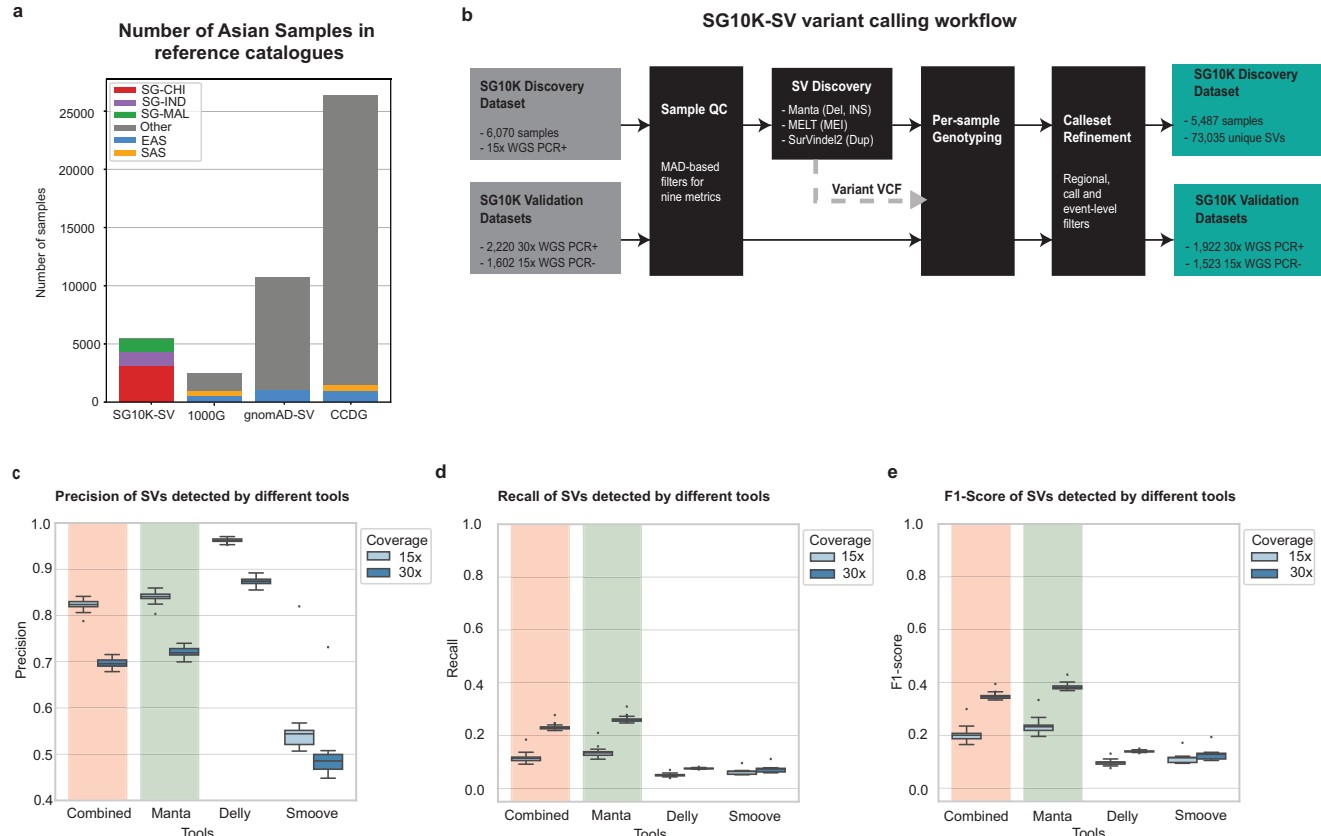

**Fig. 1 | SG10K-SV-r1.4 Structural Variant landscape. a** Number of Asian samples in SG10K-SV-r1.4 compared to (short-read derived) 1000 genomes SV, gnomAD-SV and CCDG reference studies. **b** SG10K-SV-r1.4 analysis pipeline diagram. **c−e** Benchmarking of various SV tools for SV detection using 34 1000 G samples with 2 different depths (30x coverage and downsampled 15x coverage). **c** Boxplot showing the precision between 15x and 30x coverage for each SV caller. Combined refers to variants that are detected in all three pipelines. The boxplots display the median and first/third quartiles. **d** Boxplot showing the recall between 15x and 30x coverage for each SV caller. The boxplots display the median and first/third quartiles. **e** Boxplot showing the F1-score between 15x and 30x coverage for each SV caller. The boxplots display the median and first/third quartiles.

Utilising variants in the discovery dataset, we genotyped these variants in samples from our two validation datasets to ensure that results observed in the discovery dataset are reproducible. 40,883 and 60,715 of the SVs detected in the discovery dataset were identified in the 15x PCR- and 30x PCR+ validation dataset, respectively. For the 15x_validation dataset, a total of 6775 deletions, 17,036 duplications, and 17,072 insertions were detected. In the 30x_validation dataset, 19,275 deletions, 21,377 duplications and 20,063 insertions were detected.

### SG10K_Health SV Landscape

On average, each SG10K_Health individual harboured 1439 insertions, 1584 deletions, and 1103 duplications. These figures were consistent across all three ancestries (Fig. 2c). Compared to gnomAD-SV, we detected fewer insertions and deletions per individual (insertions: 1439 in SG10K_Health vs 2612 in gnomAD-SV; Deletions: 1584 vs 3505), likely due to the higher sequencing depth (32x) of gnomAD-SV samples[10]. Confirming this hypothesis, we detected comparable insertion or deletion counts per individual in our 30x_validation dataset compared to gnomAD-SV (2030 insertions and 3200 deletions; Supplementary Fig. 6). However, despite lower sequencing depth in our discovery cohort, we detected comparable numbers of duplications compared to gnomAD-SV (1103 vs 1346), likely reflecting the improved sensitivity of the SurVIndel2 duplication-detection pipeline. Similar to previous studies[10], the majority (>70%; 53,759) of deletions, insertions and duplications were rare events with allele frequencies (AF) less than or equal to 1% (Fig. 2d and Supplementary Fig. 7). Nevertheless, we

identified 465 SVs with allele frequencies greater than 0.95 in our discovery cohort; in these cases, the reference genome bears the minor allele.

While most detected SVs were small (Fig. 2c), we identified 2678 deletions and 2065 duplications longer than 10 kb. There was a striking abundance of SVs at 300 bp, 2 kb and 6 kb (Fig. 2c). The 300 bp and 6 kb insertions corresponded to Alu and LINE1 elements respectively, the two most abundant classes of transposable elements in the human genome (-11%[34] and -17%[35] of the genome). The 2 kb SVs represent composite SVA (SINE, Variable Number Tandem Repeat, and Alu) transposons. These results highlight the pervasive contribution of repeat elements (Alu, LINE1, SVAs) in sculpting human genomic variation, and high-level similarities between our SV catalogue and other studies[12].

SVs have been reported to cluster at specific genomic regions ("hotspots"). Several factors have been proposed to influence the location of SV hotspots, such as segmental duplications and the local presence of transposable elements[36]. These factors may contribute to SV formation due to their higher propensity for DNA breakage and repair, with local transposable elements increasing the likelihood of non-allelic homologous recombination (NAHR)[37]. To identify SV hotspots, we employed hotspotter[38] (bandwidth:200,000, num.-trial=10,000, pval=$5 \times 10^{-3}$) and identified 251 regions containing higher-than-expected SV densities (Supplementary Data 5). Together, these 251 regions affected -211 Mb, in line with previous findings[29]. Notably, 36% (90 out of 251) of the hotspot regions were located within 5 Mb of the ends of the chromosomes as well as near the centromeric

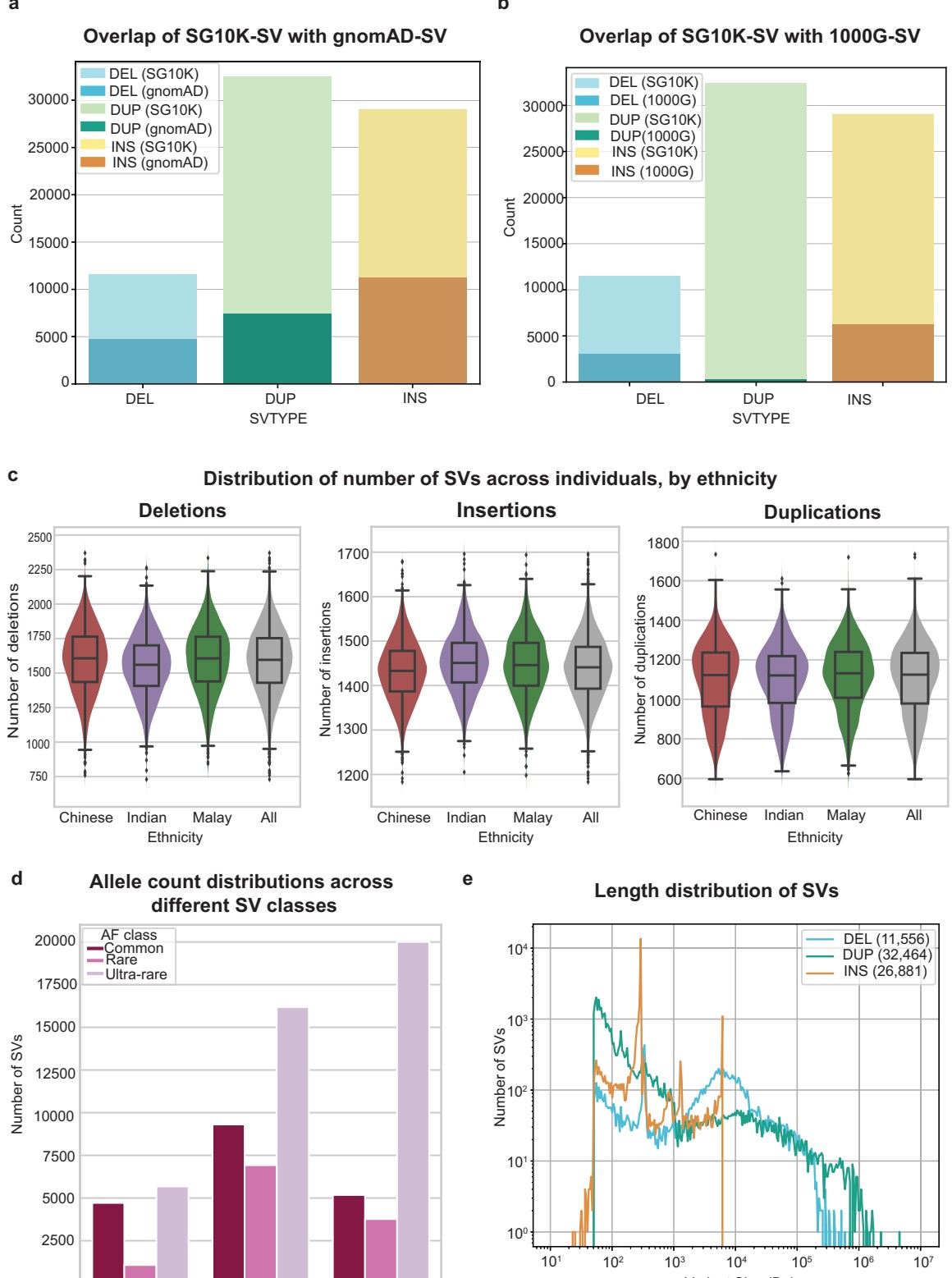

**Fig. 2 | SG10K-SV-r1.4 Structural Variant catalogue properties. a** Number of SG10K-SV-r1.4 variants detected in the discovery callset that overlap with gnomAD-SV. **b** Number of SG10K-SV-r1.4 variants detected in the discovery callset that overlap with 1000G-SV. **c** Violin plot and boxplot showing the number of SV per genome across individuals of different ethnicity group (3088 Chinese, 1237 Indians and 1144 Malays). The boxplot displays the minimum and maximum number of SVs as well as the median and the first/third quartile. DEL deletions, DUP duplications, INS insertions (including MEIs). **d** Number of SVs in different classes segregated by allele frequencies in the SG10K-SV-r1.4 discovery callset. The majority of the SVs are rare variants (AF < 1%). **e** Size distribution of SVs in SG10K-SV-r1.4 discovery callset. DEL deletions, DUP duplications, INS insertions (including MEIs). Expected Alu, SVA and LINE1 MEIs peaks at around 300 bp, 2100 bp and 6000 bp, respectively.

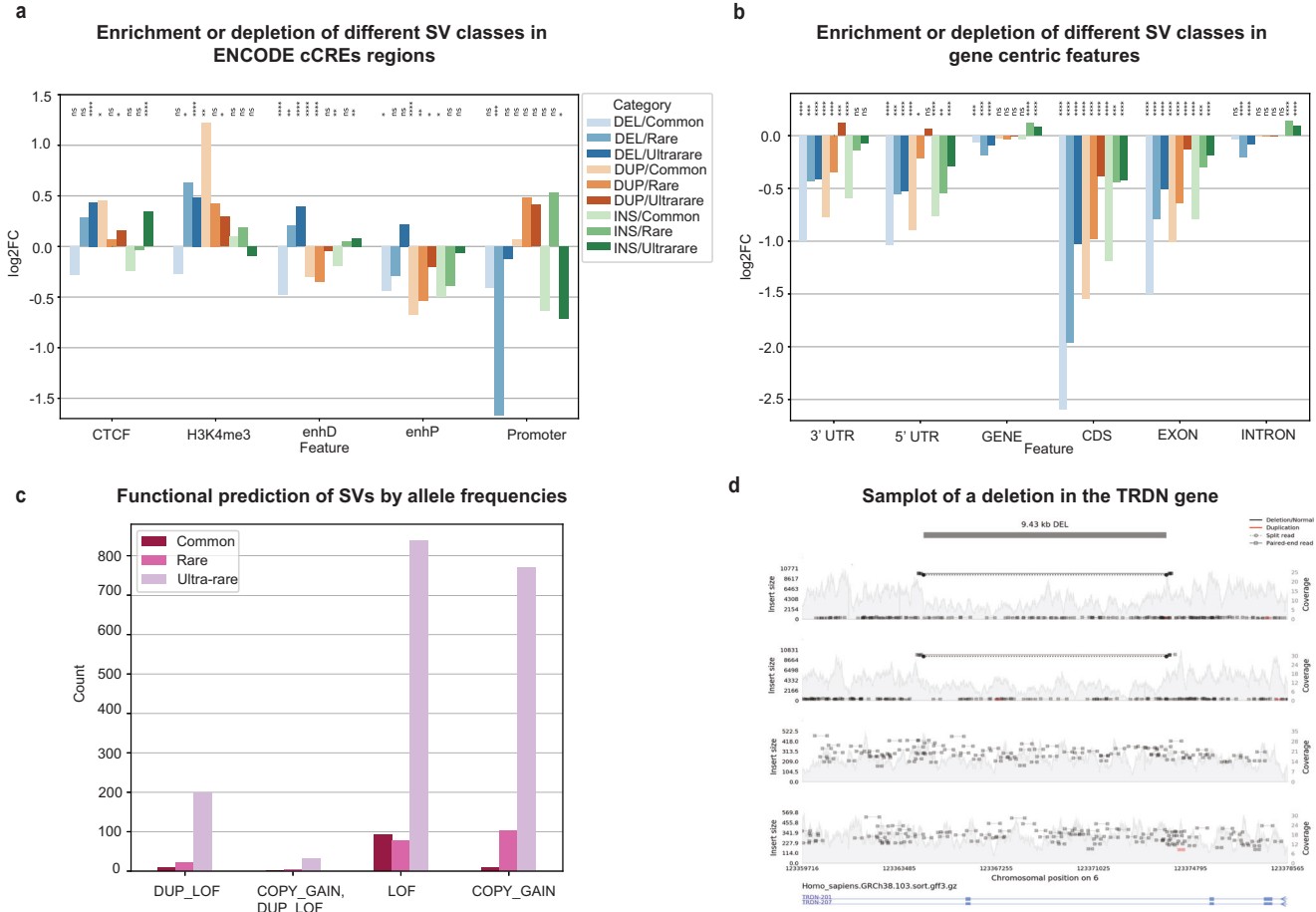

**Fig. 3 | Functional impact of structural variations in the SG10K-SV-r1.4. a:** Distribution of SVs (Deletions, Insertions, Duplications) disrupting regulatory regions (ENCODE cCREs) across allele frequency bins. Common indicates variants with allele frequency ≥0.01; rare indicates variants with allele frequency ≥0.001 and allele frequency <0.01; ultra-rare variants refers to variants with allele frequency <0.001. *P*-value was computed using 10,000 random permutations and correction with Benjamini–Hochberg false discovery rate was done. Ns indicates not significant *p*-value, * indicates *p*-value < 0.05, ***p*-value < 0.01, *** indicates *p*-value < 0.001, **** indicates *p*-value < 0.0001. The exact *p*-value for the analysis can be found in Supplementary Data 6. **b** Distribution of SVs (Deletions, Insertions, Duplications) disrupting (GENCODE) gene centric features across allele frequency bins. *p*-value was computed using 10,000 random permutations and correction with Benjamini Hochberg false discovery rate was done. Ns indicates not significant *p*-value, * indicates *p*-value < 0.05, ***p*-value < 0.01, *** indicates *p*-value < 0.001, **** indicates *p*-value < 0.0001. The exact *p*-value for the analysis can be found in Supplementary Data 7. **c** In silico prediction of functional consequences of SVs segregated by allele frequencies. **d** Samplot of a 9.43 kb deletion event overlapping the TRDN gene region.

regions. Excluding these sub-telomeric and centromeric hotspots, 88 hotspots were unique to SG10K-SV compared to gnomAD-SV. For example, we identified a 988,219 bp (chr12:124035647-125023866) hotspot region containing 58 SVs. This hotspot overlaps the *NCOR2* (Nuclear receptor corepressor 2) gene, a corepressor that is frequently altered in prostate cancer[39].

**Impact of SVs on regulatory elements and gene bodies**
To assess the impact of SG10K-SV on different categories of functional genomic regions, we overlapped the SVs with gene regulatory elements identified by ENCODE and the Epigenomics Roadmap project[40]. Regulatory elements surveyed included 926,535 putative regulatory elements annotated as distal enhancers (667,599), proximal enhancers (141,830), insulators (CTCF sites, 56,766), promoters (34,803), and poised elements (exhibiting DNase I hypersensitivity but are likely functionally gated by additional trans-acting signals), and non-promoter K4me3 regions (25,537)[40].

Common deletions (AF ≥1%) were significantly depleted at putative enhancers and insulators, consistent with a model of negative selection acting on alterations affecting gene expression (Fig. 3a). In contrast, rare (1% > AF ≥0.1%) and ultra-rare (AF < 0.1%) deletions did

not exhibit similar depletion signals. Common duplications were also significantly depleted at distal and proximal enhancers (Fig. 3a and Supplementary Data 6) again suggesting the action of purifying selection. Unexpectedly, we observed common duplications being enriched at annotated non-promoter H3K4me3 regions. To deepen this observation, we examined the intersect of 81 non-promoter H3K4me3 regions overlapping common duplications, and found that they were highly and significantly enriched for tandem repeats relative to all 25,537 H3K4me3 regions (fold enrichment: 4.6: hypergeometric *p*-value: 2.45 × 10⁻²³). We speculate that since read mapping artifacts are common at tandem repeats, it is possible that these mapping artifacts might have contributed to artefactual ChIP-seq peaks at these tandem repeat regions.

We then analysed gene bodies (UTRs, CDS, exons or introns). SVs of all three categories were strongly depleted at gene bodies, including 3'UTRs, 5'UTRs, CDS, exons, and introns (Fig. 3b and Supplementary Data 7). For example, common deletions were depleted 5-fold at coding exons, against reflecting high selection pressure on coding sequences. Similar to enhancers, rare and ultra-rare SVs showed weaker depletion patterns in exons of all types. Interestingly, intronic regions showed no deviations from background, except for a modest

elevation in rare and ultra-rare insertions. This may reflect the increased propensity of certain MEIs families to insert into the gene bodies of actively transcribed genes or GC-rich regions[41,42].

SVs deleting gene regions may cause complete or partial loss of function (LOF) effects. Conversely, duplications may lead to gene copy gain, augmenting gene dosage. Employing SVTK[43] to assess the potential impact of the SG10K SVs on protein-coding regions, we identified 2153 SVs (2.95% of 73,035) with direct predicted impact on protein coding integrity (Fig. 3c). Of these, 1008 SVs resulted in likely gene LOF. LOF-associated SVs tended to occur at low allele frequencies (AF < 1%). We identified 881 duplications predicted to cause copy number gain of one or several consecutive protein-coding genes. Copy number gain events were typically larger compared to LOF events (median size 90 kb vs 9.7 kb). These patterns are in line with findings in gnomAD-SV where the majority of protein coding affecting SVs resulted in LOF, and copy gain events exhibited larger sizes.

We assessed the potential impact of SVs on major clinically actionable genes, focusing on 81 American College of Medical Genetics and Genomics (ACMG v3.2[44]) defined actionable genes associated with highly penetrant and actionable genetic conditions. AnnotSV[45] was used to identify SVs potentially affecting at least one ACMG v3.2 gene. We found 14 SVs affecting the coding sequence integrity in 11 clinically actionable ACMG genes. For example, we identified a 9.4 kb deletion in three Chinese individuals (AF = $5.18 \times 10^{-4}$), affecting *TRDN* (Fig. 3d), encoding triadin and a key component of the calcium release complex[46]. Mutations in *TRDN* gene are associated with a recessive form of Catecholaminergic polymorphic ventricular tachycardia[47]. We also found a 9.16 kb heterozygous deletion affecting *PRKAG2* in two Chinese individuals with an AF of 0.00034 (Supplementary Fig. 8). *PRKAG2* gene encodes the gamma-2 subunit of the AMP-activated protein kinase (AMPK)[48] and mutations in *PRKAG2* gene have been associated with cardiomyopathy related to glycogen storage in heart cells[49].

## SV patterns between international cohorts

Reflecting the novelty of the SG10K-SV catalogue, 66.5% (49,601/73,035) and 86.7% (63,367/73,035) of the SVs identified were not previously reported in gnomAD-SV (Fig. 2a) or 1000G-SV catalogues (Fig. 2b), respectively. In total, 47,770 SVs in SG10K-SV did not overlap with either study (1000G-SV and gnomAD-SV). Applying a call rate cut-off across each ethnic group of ≥ 50% within SG10K-SV, we identified 42,239 SVs and hereby termed these as "Asian specific - novel" SVs. The majority of novel Asian-specific SVs identified exclusively in our catalogue exhibited lower allele frequencies than SVs identified in both SG10K-SV and gnomAD-SV or SG10K-SV and 1000G-SV (Supplementary Fig. 9).

Next, we focused on the 25,265 SVs in SG10K-SV which overlapped at least one study. We used this subset to identify SVs with a higher prevalence in Asian populations, employing fixation index (Fst)[50] analysis described in the Methods section. This resulted in identifying an additional 10,902 (out of 25,265) Asian-specific SVs.

Notable examples of Asian-specific SVs include a previously reported 2.9 kb deletion (chr2:111125617-111128520) in intron 2 of the *BIM* gene, which is associated with resistance to tyrosine kinase inhibitors[51]. This SV is present in gnomAD-SV at a higher AF in East-Asians compared to other ethnicities (AF EAS: $7.37 \times 10^{-2}$, AF others: $1.04 \times 10^{-4}$). Another example comprises a rare 19.3 kb deletion (chr16:165396_184700) spanning the *HBA1* and *HBA2* genes, associated with α-thalassemia and detected more frequently in Asian populations (AF EAS: $9.93 \times 10^{-3}$, AF others: $1.04 \times 10^{-4}$)[20].

## SVs between Asian ancestry groups

We then investigated SV patterns distinctive to the three major Asian ancestries. Principal components analysis (PCA) on the full set of SG10K-SV demonstrated ancestry-specific population clustering

(Fig. 4a), similar to SNV clustering using the SG10K_Health[23] dataset with the same samples (Supplementary Fig. 10). A similar PCA-based populations' structure organisation was found using either insertion, deletion, or duplication only events (Supplementary Fig. 11). Nevertheless, 52% of SVs were seen in only one ancestry, 13% were shared across two ancestries, and the remaining 35% of the SVs were in all three populations (Fig. 4b), supporting pervasive differences across each of these SV classes contribution to population differentiation. However, as the numbers of SVs detected as unique in a population correlated with cohort size (Supplementary Data 1) and were enriched for low-frequency SVs (Supplementary Fig. 12), it remains possible that some of these SVs may be present in other populations, but remain undetected due to their innate low allele frequency.

To gain a more granular understanding of ancestry-specific SV patterns, we calculated fixation indexes (Fst)[50] for each of the detected SV and assigned a significance score to each observation using permutation analysis (see Methods). By examining the resulting Fst trends, we found that SVs with extreme Fst values (0.7 and above) were mostly detected in small numbers of individuals (call rate <2%) not reaching significance thresholds (Fig. 4c). Amongst SVs exhibiting statistically significant Fst values, we identified 11,715 SVs displaying ancestry-specific frequency patterns, comprising 3580 deletions, 4068 insertions, and 4067 duplications (see Methods).

86 of the 11,715 ancestry-specific SVs, comprising 40 deletions, 44 duplications, and 2 insertions (Fig. 4c and Supplementary Data 8) resulted in gene copy gains or LOFs. These gene integrity affecting ancestry-specific SVs were observed across a range of population-level allele frequencies (Fig. 4d) and included several previously reported SVs. For example, we observed a 2.7 kb deletion (chr6:8432262-8434992) in the *SLC35B3* gene, involved in the transport of 3'-phosphoadenosine-5'-phosphosulfate (PAPS)[52]. This SV was common (AF = 0.016) in East Asians within the gnomAD-SV catalogue and rare in other ancestries. This SV also exhibited significantly higher AFs in Chinese and Malays compared to Indians (AF SG-Chinese: 0.0149, AF SG-Indian: 0.0018, AF SG-Malay: 0.0196). Another example was a rare 8.8 kb deletion (chr6:158745097-158753965) overlapping *STYL3*, present at low allele frequency (AF = 0.00015) in gnomAD-SV. This SV shows a higher allele frequency in East Asians (AF = 0.00148) compared to other ancestries within gnomAD-SV and was not observed in individuals of South Asian ancestry in gnomAD-SV. We observed a similar allele frequency for Chinese and Indians (AF SG-Chinese: 0.0012, AF SG-Indian: 0). However, this deletion appears to be common among individuals of Malay ethnicity (AF SG-Malay: 0.019). A third example was a 9.8 kb duplication overlapping *PROCR*, encoding a receptor for activated protein C[53]. This duplication was seen only in Asians in gnomAD-SV (EAS AF = 0.008; SAS AF = 0.0002). This SV also exhibited a higher allele frequency in Chinese and Malays compared to Indians (AF SG-Chinese: 0.0149, AF SG-Indian: 0, AF SG-Malay: 0.0026). Finally, we identified a 18 kb duplication (chr6:73747426-73766255) overlapping *CD109*, a glycosylphosphatidylinositol (GPI) anchored protein[54], and increased *CD109* gene expression has been observed in several cancers[55,56]. This duplication was observed in individuals of South Asian ancestry but not in East Asians (EAS AF = 0, SAS AF = 0.002). We noted a similar trend in allele frequencies within our catalogue (AF SG-Chinese: $3.24 \times 10^{-4}$, AF SG-Indian: 0.024, AF SG-Malay: 0.0017).

Importantly, we also discovered previously unreported SVs. One such SV was a 942 bp duplication overlapping *SMC1B*, encoding a protein involved in chromatid cohesion and DNA recombination during meiosis and mitosis[57]. The AF of this SV was higher in Chinese and Indians compared to Malays (AF SG-Chinese: 0.28, AF SG-Indian: 0.25, AF SG-Malay: 0.15). We also detected a 84 bp deletion in *ZNF83*, and missense point mutations in ZNF83 have been associated with poor prognosis in urothelial carcinoma[58]. This event was detected with the lowest AF in Indians (AF SG-Chinese: 0.49, AF SG-Indian: 0.17,

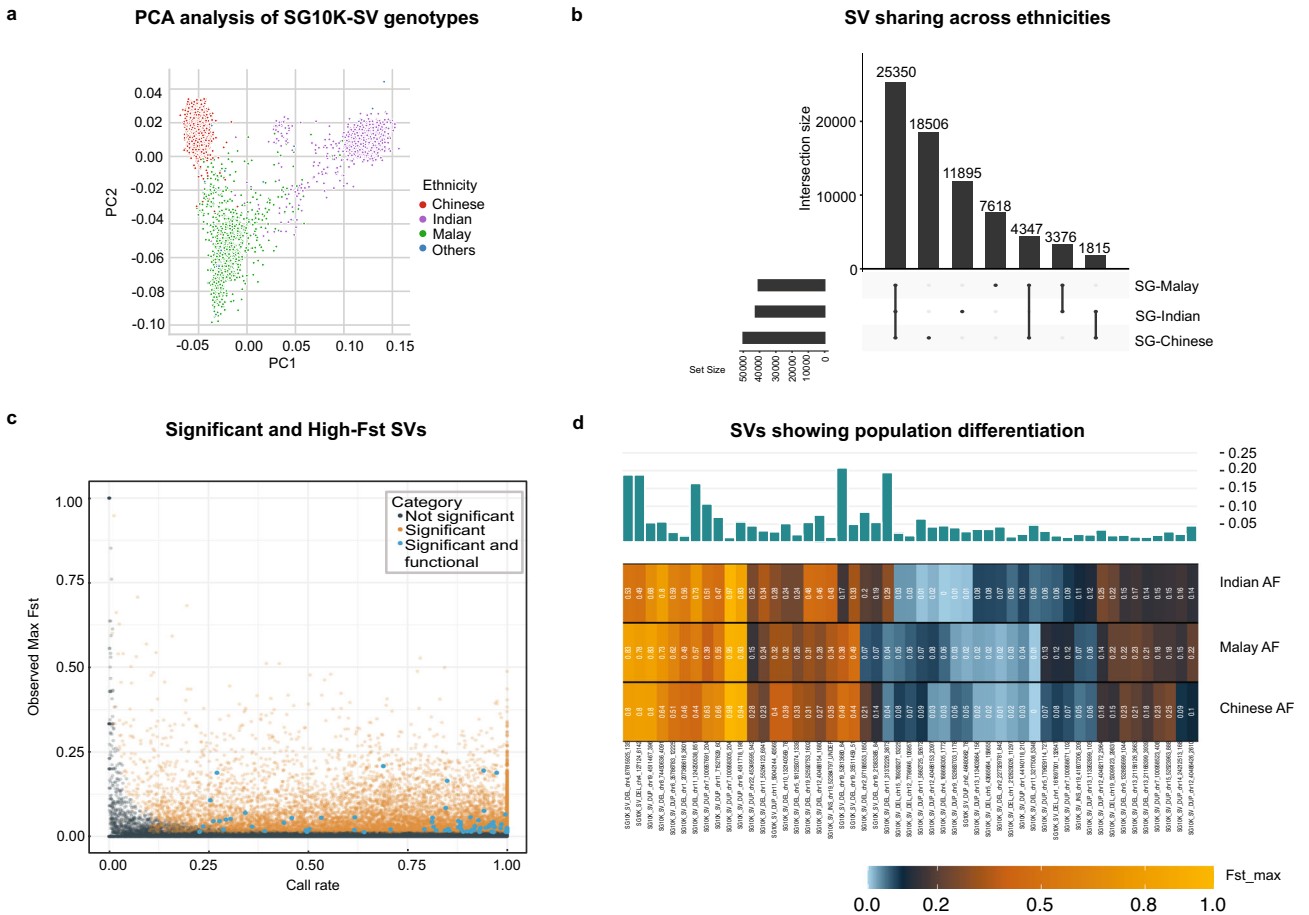

**Fig. 4 | Population specificity of SVs. a** Population structure revealed by PCA analysis of SG10K-SV-r1.4 genotype values. Each point corresponds to an individual, coloured according to its ethnicity, x and y axis represents the first two principal component respectively. **b** Proportion of SVs found in all, two or one populations. **c** Scatter plot of SV's fixation index (Fst) as a function of their call rate. **d** Allele frequencies in Chinese, Indian and Malay for selected SVs with elevated fixation index (Fst).

AF SG-Malay: 0.38). We found an Indian-specific insertion included a 209 bp SV overlapping *CEACAM3* (AF SG-Chinese: 0.05, AF SG-Indian: 0.11, AF SG-Malay: 0.07), a cell adhesion molecule that plays a crucial role in the innate immune response to bacterial infections[59]. Finally, we identified a 6.9 kb deletion overlapping *TRIM48*, a gene predicted to encode protein that function as E3 ubiquitin ligases and has been shown to promote *ASK1* activation[60]. The AF of this deletion was higher in Indian compared to Malay and Chinese individuals (AF SG-Chinese: 0.23, AF SG-Indian: 0.34, AF SG-Malay: 0.24). Collectively, our analyses demonstrate that numerous population-specific SVs among Asians can be detected using SG10K-SV.

### SVs exhibit *cis*-linkage to disease GWAS loci

Finally, SVs are gaining prominence as potential genetic drivers of disease susceptibility, drug response and other phenotypes[61]. To explore potential associations between SVs and biological phenotypes, we hypothesised that certain trait-associated lead SNPs identified by GWAS (GWAS-lead SNPs) might not (and often do not) represent the actual causative variant. Conventional GWAS analysis thus often requires pinpointing underlying causal variants using fine-scale genetic mapping to assess variants showing high linkage disequilibrium (LD) with lead SNPs. Since SVs are large variants in terms of genomic span, it is possible that certain SVs in strong LD with GWAS lead SNPs might be the causative genetic alteration[62].

To explore this possibility, we performed LD analysis between SG10K-SVs and previously reported WGS-inferred SG10K_Health small variants (SNPs or short indels)[23]. LD was computed for high-confidence (call rate ≥ 0.8) common (MAF ≥ 1%) SVs ($n = 6772$) and small variants ($n = 9,450,184$) located within a 1 Mb distance (Fig. 5a). 15.8% of SVs were not in LD with any SG10K_Health small variants ($R^2 < 0.2$), suggesting that a substantial proportion of SVs represents genetic variability that might be overlooked in conventional genetic association analyses. 3909 of the 6772 high-confidence common SVs were in strong LD with 164,992 SG10K_Health SNPs ($R^2 \geq 0.8$). Of these, 885 SVs were in strong LD with 2151 SG10K_Health SNPs matching lead SNPs from the EBI GWAS catalogue[63] based on genomic positions, with 385 SVs (154 deletions, 34 duplications, and 197 insertions) in strong LD with 664 lead SNPs from GWAS focused on Asian cohorts. Supplementary Data 9 lists all 385 SG10K-SVs candidate causative genetic alteration together with their associated GWAS lead SNPs.

GWAS-lead SNPs are often found in non-coding regions of the genome. Our analysis highlighted two exonic-associated SVs in high LD with these non-exonic SNPs, where the former may represent underlying causal variants. We focused on the subset of exon-overlapping SG10K-SVs, since they could most directly be assigned a functional consequence. A first example include a predicted LOF inducing SV deletion (chr1: 152583066-152615264) which overlaps *LCE3B* and *LCE3C* genes. This SV was in strong LD with two GWAS lead SNPs (rs4085613 ($R^2 = 0.97$) and rs4845459 ($R^2 = 0.98$); Fig. 5b)[64,65]. Notably, both SNPs are associated with psoriasis ($P = 7 \times 10^{-30}$ and $P = 6 \times 10^{-11}$) in individuals of East Asian ancestries. Both SNPs are not found in the coding region of the genes and hence, our analysis suggests that the linked LOF SV should also be considered a potential causal variant for psoriasis in this locus.

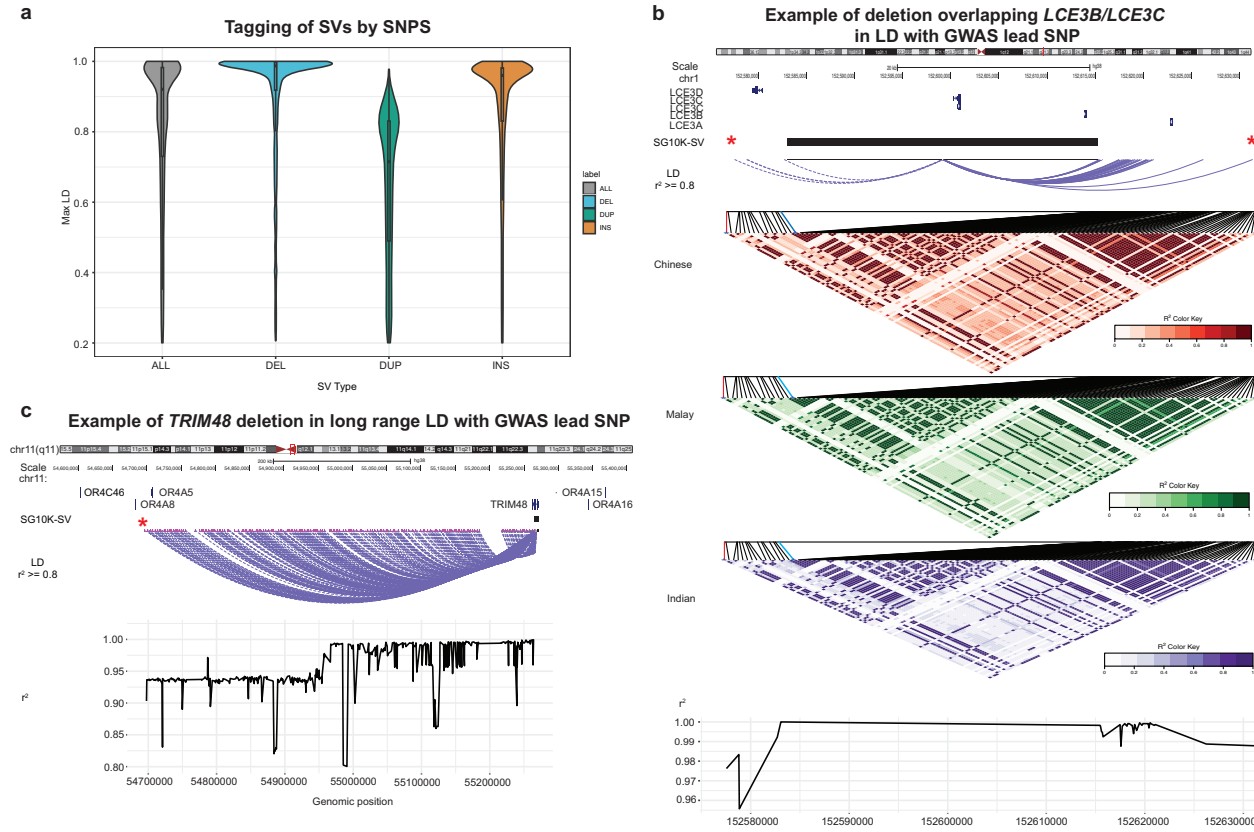

**Fig. 5 | Linkage disequilibrium between SVs and SNPs. a** Tagging of SVs by SNPs: Violin plots and boxplots showing the distribution of the maximum $R^2$ value to SNPs for each SV. The boxplots and violin plots were plotted for 1400 deletions, 1394 duplications and 2903 insertions, which are in LD with SNPs/small indels in the SG10K_Health dataset. The boxplots display the median and first/third quartiles. **b** Candidate causal SV: Example of a deletion affecting *LCE3B/LCE3C* gene, in high LD with two Psoriasis GWAS SNPs. The SNPs are significantly associated with Psoriasis. LD structure plots are shown for the three ethnicities. The star indicates the GWAS lead SNP and the black bar indicates the SV. The line plot shows the r2 of variants in the region with respect to the SV. **c** Candidate causal SV: Example of a deletion in *TRIM48* gene, in high LD with an intergenic GWAS-lead SNP associated with altered glomerular filtration rate. The lines indicate LD between GWAS-lead SNP and deletion with r2 >= 0.8. The star indicates the GWAS lead SNP and the black bar indicates the SV. The line plot shows the r2 of variants in the region with respect to the SV. Genomic region overviews in panels **b** and **c** include[84] screenshots from http://genome.ucsc.edu.

We also observed a predicted LOF SV (chr11:55,264,123-55,271,064) deleting exons 2 to 6 of *TRIM48* exhibited strong LD ($R^2 = 0.903$; Fig. 5c) with an intergenic GWAS-lead SNP (chr11:54,697,371; rs11532186) associated with altered filtration rate. Notably, an integrative analysis of genetic association and gene expression in a cohort of patients with reduced kidney function identified *TRIM48* among the top causal candidates for urine metabolite variation[66,67]. These examples support the value of including SG10K-SVs in analyses of genetic drivers of phenotypic variation in Asian cohorts. The full list of SVs in high LD with GWAS lead SNPs is reported in Supplementary Data 9.

## Discussion

We generated a comprehensive catalogue of SVs in 8392 Singaporeans containing 73,035 SVs. Compared to previous studies analysing primarily populations of Eurocentric ancestry, our samples enabled us to assess patterns of SV genetic diversity across Asia, leveraging on Singapore as a diverse multi-ancestry community. In particular, little is known about the SV landscape in Malay individuals. Malays are the third largest ethnic group in Asia. While the majority of the 220 million individuals of Malay ethnicity resides in Indonesia and Malaysia, they are geographically distributed across several countries in Southeast Asia, including Singapore and Sri Lanka[68]. Previously, Wu et al.[7] investigated the population structure of the three Singaporean populations with the 1000 Genomes project populations using small

variants and reported an ancestral component that is largely specific to the Malays in Singapore[7]. This result indicates the importance of including individuals of Malay ethnicity in large-scale population-based SV studies so as to uncover SVs unique to this community. Previous studies have characterised SVs in the Chinese (East Asian) and Indian (South Asian) populations[10,12]. However, our current study provides a much more comprehensive catalogue of SVs for these populations by analysing a significantly larger number of samples than previous efforts. Overall, our findings reiterate the importance of creating a comprehensive population-specific database of SVs to fill the gap of our understanding of genetic diversity in Asian populations.

While clearly a first-generation catalogue, the SG10K-SV database identified novel SVs that were not seen in existing population-based SV catalogues, such as gnomAD-SV and 1000G-SV. We identified 11,715 SVs displaying ancestry-specific frequency patterns, of which, 86 SVs had functional implications. These 86 SVs included SVs that were reported previously in Asian population as well as novel SVs showing differences in allele frequencies between the ethnic groups. For example, we identified a duplication overlapping *CD109* that was also seen in individuals of South Asian ancestry in gnomAD-SV. We observed a similar allele frequency trend in SG10K-SV for this SV. We also identified a rare deletion in *STYL3* gene that was also present in East Asians in the gnomAD-SV catalogue. However, using the SG10K-SV catalogue, we observed that this deletion is common among individuals of Malay ethnicity with an allele frequency of 0.019. Apart from

known SVs, we also identified multiple novel SVs showing significant differences in allele frequencies between the ancestry groups. We identified a novel duplication overlapping *SMC1B* that has a higher AF in Chinese and Indians compared to Malays. We also detected another novel Indian-specific insertion overlapping *CEACAM3*. These findings reiterate the importance of creating a population-specific SV catalogue which allows us to understand the genetic variations that drive differences between the ethnic groups.

The SG10K-SV catalogue has also enabled us to identify potential SVs associated with phenotypic variations. Beyond SVs affecting gene function, integrating SG10K-SVs with SG10K_Health SNPs enabled us to identify LD patterns between polymorphic single nucleotide (SNPs) and large-scale genomic variations (SVs). Integrating the SG10k-SV and SG10K_Health SNPs collections with Asian cohorts catalogued GWAS lead SNPs, we were able to identify potential causative SVs plausibly associated with disease phenotypes. For example, we identified a LOF deletion affecting *LCE3B/LC23C* gene that is in strong LD with two non-coding GWAS lead SNPs associated with Psoriasis. This demonstrates the value of the SG10K-SV database, which allowed the identification of potential causative SVs that are in strong LD with GWAS lead SNPs associated with disease phenotypes in the Asian cohort.

Our study has several limitations. SV discovery is challenging, and the full spectrum of SVs in the human genome remains poorly understood. The findings presented here are primarily derived from 15x short-read WGS, and thus clearly underpowered both in terms of sequencing read length and sequence coverage to capture all possible SVs present in the Asian population. Existing algorithms rely on sequencing coverage and split-reads from the short-read WGS data to detect SVs, and hence, the precise identification of genomic coordinates and length of tandem duplicates and large insertions is hampered. In addition, at present, our SV callers captured only the three most commonly analysed SVs (deletions, insertions and duplications), but did not consider other SV classes (inversions, translocation) that are also present in human genomes and are likely to have biological consequences. Using long-read sequencing in the near future, either as a single modality or coupled with high-coverage short-read sequencing, will allow us to identify substantially more SVs, clarify SVs in repetitive regions, and define new classes of SVs. Notwithstanding these shortcomings, the SG10K-SV dataset is, by far, the largest Asian SV database. This resource will be valuable to understanding the genetic diversity of the Asian population and how these variations underpin health and disease in the Asian population.

## Methods

### WGS data quality control

We processed WGS data collected from the SG10K_Health[23] study. SG10K Health comprises alignments and variant calls for SNVs and INDELs from 9 local cohorts, including 9770 healthy individuals. Data generation involved WGS of blood DNA samples (Illumina short-reads) and subsequent analysis following GATK best practices (GATK4 GRCh38)[69] to generate individual sample level CRAM files. It also included QC checks intended to discard samples with poor sequencing quality (e.g. hard filters for error rate and contamination), unusual numbers of calls (e.g. Median absolute deviation (MAD)-based filters on het/hom ratio), chromosome aneuploidies, and/or samples with related individuals in the same cohort (see methods in Wong et al., 2023 for additional details).

Using an in-house developed pipeline, we calculated the coverage, alignment and GC bias metrics from the SG10K Health CRAMs. In total, nine metrics were considered for downstream filtering, chosen to represent the type of evidence used by SV calling algorithms:

- median autosome coverage: The median coverage in autosomes, excluding (i) bases in reads with low mapping quality (mapq <20); (ii) bases in reads marked as PCR duplicates, and (iii) overlapping bases in read pairs; calculated with mosdepth[70].

- mad autosome coverage: The median absolute deviation of coverage in autosomes after coverage filters are applied (see "median autosome coverage"); calculated with mosdepth[70].
- pct autosomes 1x: The percentage of bases that attained at least 1X sequence coverage in autosomes, after coverage filters are applied (see median autosome coverage); calculated with mosdepth[70].
- pct reads aligned: The percentage of PF reads that align to the reference; calculated with picard AlignmentSummaryMetrics[71]. PF reads refer to reads that passes Illumina's filter.
- pct reads properly paired: The percentage of reads that align as proper pairs as calculated with samtools stats[72]. Properly pair reads are reads in which both reads in the pair are mapped and they are mapped within the range from each other based on the estimated insert size distribution.
- median insert size: The median insert size of aligned reads; calculated with picard InsertSizeMetrics[71].
- mad insert size: The median absolute deviation of insert sizes; calculated with picard InsertSizeMetrics[71].
- gc dropout: Illumina-style GC dropout metric; calculated with picard GcBiasSummaryMetrics[71].
- at dropout: Illumina-style AT dropout metric; calculated with picard GcBiasSummaryMetrics[71].

In each cohort, we discarded samples outside 8 MAD from the median for at least one of the nine metrics considered. Such filters led to the exclusion of 1378 samples, thus leaving 8392 samples for downstream analysis.

### Deletions and insertions detection

In this study, we employed Manta[26] to identify deletions and insertions separately in single samples, followed by SVimmer[73] to obtain a putative cohort-wide consensus set. Individual-level genotype calls within this uniformly-defined discovery SV set were then refined using Graphtyper2[74].

Manta v1.6 was executed in the single sample mode to identify deletions and insertions in the discovery dataset. We used the default parameters and further filtered the single-sample VCF to retain (i) calls that pass filters, (ii) with a length of 50 bp or more and (iii) of the selected variant types (deletions and insertions).

SV discovery from short read data is notedly a challenging task[75]. Moreover, since the majority of our dataset consists of 15x genomes (Supplementary Data 1), we expect lower sensitivity compared to what has been reported in higher-depth studies[10]. In order to overcome these limitations, we have incorporated additional clustering and re-genotyping steps, which are known to improve detection power in short-read-based studies. In brief, the goal is to aggregate all SV candidates identified when evaluating each sample individually (SV clustering), and then re-assess the original data for the presence/absence of these calls (SV re-genotyping).

Prior to clustering, we sought to discard any samples that displayed an unusual number of calls for any of the SV types considered, by applying an 8-MAD filter on a per-cohort basis, analogous to the strategy previously used during sample QC. For Manta, no samples were discarded after applying such a filter, suggesting that the upstream sample QC is already adequate to flag unusual samples. We then clustered SV candidates in each of the call sets obtained during the discovery step using svimmer[73], which we ran with default parameters to aggregate events across all samples in the discovery dataset. We then performed re-genotyping for each sample using Graphtyper[74] v2.5.1 with default parameters and merged the individual genotype VCF across all samples using vcf_merge subcommand within Graphtyper2. We then set all genotypes that were marked by Graphtyper2 as Fail to null using Hail[76].

We retained SVs that passed the following criteria: (1) filter = PASS; (2) SVMODEL = AGGREGATED; (3) SVTYPE = INS or SVTYPE = DEL; (4) SVs with length ≥ 50 and SVs with length ≤ 1,000,000; (5) SVs that are polymorphic and has at least 1 sample with a homozygous reference genotype.

In order to create a high-confidence SV dataset, we applied additional filters recommended by Graphtyper2. For deletions, we retained variants that passed the following criteria: (1) ABHet > 0.30 | ABHet <0; (2) AC/NUM_MERGED_SVS < 25; (3) PASS_AC > 0; (4) PASS_ratio > 0.1 and lastly (5) QD > 12. For insertions, we retained variants that passed the following criteria: (1) ABHet > 0.25 | ABHet <0; (2) AC/NUM_MERGED_SVS < 25; (3) PASS_AC > 0; (4) PASS_ratio > 0.1 and (5) MaxAAS > 4.

## Mobile element insertions detection

MELT v2.2.2[33] was executed using MELT-Split with default parameters in a four-step process to identify different classes of MEIs (Alu, SVA, LINE1) in the SG10K-SV discovery set. First, MELT-indivAnalysis was used to identify MEI in each sample. Second, MELT-groupAnalysis was used to aggregate MEIs across all samples in the discovery dataset. Third, we performed re-genotyping for each sample using the merged MEI information obtained from step 2 using Genotype feature in MELT. Lastly, MELT-Split uses the MELT-makeVCF function to filter and merge MEIs information across all samples into a single VCF. The four-step MEI discovery was run separately for each MEI class. We extract only variants that PASS the filters indicated by MELT for downstream analysis.

For the two validation datasets, we used the output file from MELT-groupAnalysis, which contains aggregated MEIs across all samples in the discovery dataset, to re-genotype MEIs in each sample in the two validation datasets. Lastly, we used the MELT-makeVCF function to filter and merge MEIs across all samples into a single VCF. We extract only variants that PASS the filters indicated by MELT and polymorphic variants with at least one 1 sample having homozygous reference genotype.

## Duplications detection

We ran SurVIndel2[32] with default parameters on each sample in the discovery set and only retained tandem duplications. Duplications were left-aligned using the normalised utility in SurVIndel2. Then, we clustered the duplications as recommended in the manuscript of SurVIndel2, merging events whose length differ by less than 100 bp, and whose extremities were located within 200 bp of each other in a manner analogous to that employed by SVimmer for insertion and deletion clustering.

Next, we used the companion re-genotyper of SurVIndel2, SurV-Typer, to genotype each duplication in each sample. The genotyped duplications for each sample were merged using bcftools merge. Duphold[77] was ran on the calls generated by SurVIndel2. We set genotypes calls to PASS if they meet the following criteria: if genotype is (1) homozygous reference and FT == PASS; (2) heterozygous and FT == PASS and DHBFC > 1.3; (3) homozygous alternate and FT == PASS and DHBFC > 1.3. Genotypes that failed these criteria were set to Null using Hail. Lastly, we retained duplications that passed the following criteria: (1) duplications with length ≥ 50 and duplications with length ≤ 1,000,000; (2) SVs that are polymorphic and has at least 1 sample with a homozygous reference genotype.

## Callset refinement and merging of individual variant callset into SG10K-SV Release 1.4 discovery and validation datasets

For the last step of the SV pipeline, we used a combination of regional, call and event-specific filters to further refine the outputs of the re-genotyping step, aiming to reduce the number of false positives in our dataset. Region-specific filters were applied consistently across all samples before generating the final SG10K-SV release 1.4 to (i) retain

events in autosomal contigs (chr1-22), (ii) exclude those that occur in centromeres, telomeres, heterochromatin region[27], (iii) exclude regions in the primary assembly that overlap with ALT contigs and (iv) exclude N-masked regions of the reference genome.

## Benchmarking of tools for duplication calling

Benchmarking structural variations (SVs) generated by short-read methods is often done using long-read-based ground truth catalogues. The Human Genome SV Consortium (HGSVC) released HGSVC2, a comprehensive set of SVs detected in 35 samples in the 1000 Genome Project using PacBio HiFi and CLR reads[29]. Additionally, CRAM files at 30x coverage are available for all the samples[78]. We used 10 samples for our benchmarking effort. We down-sampled these 10 samples to a sequencing depth of 15x using samtools[72] to mimic our discovery set. Next, we ran our pipeline on a dataset comprising 5487 discovery samples plus the 10 benchmarking samples. Finally, we obtained a call set for each sample by retaining SVs with an allele count of at least 1 and an FS value of PASS. We used an in-house tool (https://github.com/Mesh89/SVComparator) to compare, for each sample, the predicted SVs with the set of SVs reported in HGSVC2. Our pipeline reports tandem duplications and insertions separately, while HGSVC2 only reports deletions and insertions; tandem duplications are considered insertions. For this reason, we could not measure the sensitivity of our duplications and insertions separately.

## Principal component analysis

To investigate the relationship between the different ethnic groups in Singapore, we performed principal component analysis (PCA) using all variants (deletions, insertions, duplications and MEIs) genotypes using the "hl.hwe_normalized_pca()" function in Hail[76]. We performed PCA on all samples in the discovery dataset. The results indicate that PC1 and PC2 can segregate the individuals by their ethnic groups. We also performed PCA on all samples in the discovery dataset for each variant type separately. The results obtained per variant type recapitulated the population structure when all variants were analysed together.

## Comparison of the number of Asian samples across different population-based SV studies

We obtain the ancestry composition of 3 major studies with SV, namely 1) gnomAD-SV[10] 2) 1000 Genomes Project (1KG)[12] 3) Centers for Common Disease Genomics (CCDG)[8]. Samples in gnomAD-SV were grouped into EAS (gnomAD-SV East Asian (EAS) sample) and Other (all other non-EAS sample), while 1KG was grouped into EAS (1KG's sample found in superpopulation of East Asian ancestry (EAS)), SAS (1KG's sample found in superpopulation of South Asian ancestry (SAS)) and Other (1KG's superpopulation which are not EAS and SAS) and CCDG was grouped into EAS (CCDG's sample of EAS ancestry), SAS (CCDG's sample of SAS ancestry) and Other (CCDG's sample of non-EAS or non-SAS ancestry). SG10K-SV's sample were grouped into SG-CHI (individuals of self-reported "Chinese" ethnicity), SG-MAL (individuals of self-reported Malay ethnicity) and SG-IND (individuals of self-reported Indian ethnicity). Sample count of each group was plotted in a stacked barplot for each project.

## Comparison to SVs from gnomAD-SV

We obtained the hg38 lift-over gnomAD-SV callset from NCBI's dbvar study "nstd166".

The dataset can be obtained from https://ftp.ncbi.nlm.nih.gov/pub/dbVar/data/Homo_sapiens/by_study/vcf/nstd166.GRCh38.variant_call.vcf.gz. We considered any SG10K-SV to be novel if no overlapping gnomAD-SV could be identified using a approach similar to our SVimmer-based clustering of individual sample derived SV candidates, aggregating events across gnomAD-SV and SG10K-SV with SVimmer[73] default parameters.

## Comparison to SVs from 1000G-SV

We obtained the VCF contain the SV calls from 1000G-SV from: https://ftp.1000genomes.ebi.ac.uk/vol1/ftp/phase3/integrated_sv_map/supporting/GRCh38_positions/ALL.wgs.mergedSV.v8.20130502.svs.genotypes.GRCh38.vcf.gz.

We considered any SG10K-SV to be novel if the SV does not overlap 1000G-SV data using an approach similar to our SVimmer-based clustering of individual sample derived SV candidates using SVimmer[73] default parameters.

## Enrichment analysis

To calculate the relative enrichment for genic and non-coding regions of the genome, we downloaded the ENCODE cCRE track[79] and GENCODE v40[80] annotation from UCSC table browser.

First, we partitioned the SG10K-SV dataset into three groups (ultra-rare, rare and common) based on the allele frequency of the variants using bcftools[72] (version 1.16) filter function. Ultra-rare variants are variants with AF < 0.001; rare variants are variants with AF >= 0.001 and AF < 0.01 and lastly, common variants are variants with AF >= 0.01. The partitioned VCF files were transformed into bed files with bcftools query and a custom script. To calculate the relative enrichment of SVs in non-coding cCRE regions, we retain only variants that do not overlap any exons using bedtools[81] (v2.30.0) intersect. Next, we count the number of variants which overlaps cCRE regions and genic regions using bedtools intersect. Lastly, we performed permutation tests for the different cCRE regulatory elements or genic regions that overlap SVs. For the permutation tests, the null distribution is calculated by the number of overlaps between cCRE regulatory elements or genic regions and randomly shuffled SV locations. We generated 10,000 random SV sets constraining the coordinates of the shuffled SVs within the same chromosome and non-overlapping. The enrichment of a specific cCRE regulatory elements or gene region and SV overlap is expressed as the log2 fold change of the number of actual SVs that overlap the specific regulatory or gene regions divided by the average of the null distribution. Each cCRE or genic region which has 0 overlap with shuffled SV are assigned an arbitrary count of 1 to prevent mathematical error. A positive log2 fold change indicates an enrichment of SVs in the specific regulatory or gene region compared to a random null distribution, whereas a negative log2 fold change indicates a depletion of SVs in the specific regulatory or gene region when compared against the null distribution. Lastly, the $p$-value was calculated as follows and corrected with Benjamini-Horchberg False Discovery Rate with the scipy.stats (v1.11.4) package:

$$p-value = \frac{[Number\ of\ times\ abs(\log 2\ simulated\ fold\ change) >= abs(\log 2\ fold\ change\ actual)]}{10,000}$$

(1)

## SV annotations

We annotated the SV VCF using SVTK[43] with default parameters to associated SVs with GENCODE release 40 genes and transcripts using the following command:

svtk annotate --gencode -/gencode_v40/gencode.v40.primary_assembly.annotation.gtf SG10K-SV-Release-1.4-HighConfidenceSV-withMetadata.vcf.gz SG10K-SV-Release-1.4-HighConfidenceSV-withMetadata.svtk.gencode40.vcf

We focused on SVs that were annotated as loss of function (LOF), copy gain, duplications LOF (DUP_LOF). A deletion is predicted as LOF when it overlap at least one exon of a gene. A duplication is predicted as LOF when both the start and end of the duplication are contained within the exon of a gene. On the other hand, a duplication is annotated as DUP_LOF if a duplication overlaps at least one exon of a gene. A duplication is annotated as copy gain if it spans the entire gene.

Lastly, an insertion is predicted as a LOF if a sequence is inserted into an exon.

To identify SVs affecting medically relevant genes, we annotated the SG10K-SV VCF using AnnotSV v3.4[45] with default parameters to identify SVs overlapping with the genes listed in ACMG version 3.2[44].

AnnotSV -SVinputFile SG10K-SV-Release-1.4-HighConfidenceSV-withMetadata.variantsonly.bed -svtBEDcol 4 -outputdir AnnotSV -genomeBuild GRCh38 -bcftools /bin/bcftools -bedtools /bin/bedtools -annotationsDir /usr/local/share/AnnotSV/

## Identifying hotspots in SG10K-SV Release 1.4

To identify SV hotspot in the SG10K-SV dataset and gnomAD-SV dataset, we employed hotspotter from the primatR package[38] with the following parameters: (bandwidth:200,000, num.trial=10,000, pval=$5 \times 10^{-3}$). To identify hotspots unique to our dataset, we used bedtools[81] intersect with the "-v" function to find hotspot regions that are absent in gnomAD-SV.

## Linkage disequilibrium analysis between SNPs and SVs

To explore the relationship between SVs and SNPs, we conducted pairwise linkage disequilibrium (LD) analysis between each SV and small variants identified in SG10K_Health[23]. We compute LD between high-confidence (call rate $\geq$ 0.8) common (MAF $\geq$ 1%) SVs ($n = 6772$) and small variants ($n = 9,450,184$) located within a 1 Mb window using PLINK v1.9[82], similar to the approach used by TOPMED[83].

Known GWAS lead SNPs were retrieved from the NHGRI-EBI GWAS catalogue v1.0.2, only studies where the GWAS summary init-sample information detailing the studied cohort(s) composition contains, in part, or in full, the token chinese, Chinese, Japan, japan, Asian, asian, Asia, asia, Korea, korea, Taiwan, taiwan, Malay, malay, India or india, involving Asian individuals containing cohorts were retained. Finally, we found SNPs in common between the filtered NHGRI-EBI GWAS catalogue and SG10K-SNP that were in high LD ($R^2 \geq 0.8$) with an SV in SG10K-SV.

## Fixation index (Fst) calculation

We computed Fst values using the hudson_fst function from the scikit allel Python package. The calculation involved comparing allele frequencies (AF) between pairs of populations. For SG10K-SV, we performed three pairwise comparisons, 1) Chinese vs. Indian, 2) Chinese vs. Malay, and 3) Indian vs. Malay populations. The resulting Fst values were obtained for each pair and the maximum Fst value was kept for each SG10K-SV event along with the annotation of which pair-wise comparison generated the Fst value. Next, to assign $p$-values to each Fst value, we conducted permutation analysis. This approach involved maintaining the original genotype matrix while randomly shuffling the ancestry labels across 1000 iterations, for each of which the Fst was recalculated. The significance of the observed Fst values was then determined by comparing these against the distribution of Fst values obtained from the permuted data, calculating a $p$-value based on the proportion of permuted Fst values lower than the observed value. FDR was applied to adjust for multiple testing. Subsequently, we applied additional filtering on the obtained FDR values to identify SVs with significant Fst. Specifically, we focused on events with an FDR threshold of less than 1% and an Fst value greater than the mean of the entire dataset.

For gnomAD-SV Fst calculation, we compared EAS versus the non-EAS ancestry group using the VCF downloaded from https://ftp.ncbi.nlm.nih.gov/pub/dbVar/data/Homo_sapiens/by_study/genotype/nstd166/gnomad_v2.1_sv.sites.accessioned.vcf.gz which contains the necessary tags of the ancestry group's allele call type, for example the EAS_N_HOMREF, EAS_N_HET and EAS_N_HOMALT tags representing East Asian's number of sample called homozygous reference (hom_ref), heterozygous (het) and homozygous alternate (hom_alt) allele respectively. With the count for each

allele call type, we generated a GenotypeArray in scikit allel package with each element in the GenotypeArray being genotype status, [0,0] for hom_ref, [0,1] for het and [1,1] for hom_alt, based on the count of EAS ancestry group allele call type and a similar GenotypeArray was produced for the non-EAS ancestry group. The EAS GenotypeArray and non-EAS GenotypeArray was used to calculate the allele count for the 2 group with count_alleles function and the generated allele count used to calculate Fst with hudson_fst function.

To generate the *p*-value for gnomAD-SV, we combine the EAS GenotypeArray and non-EAS GenotypeArray and noted the length, n, of EAS GenotypeArray, we then shuffle the combined GenotypeArray, then split the shuffled GenotypeArray into shuffled EAS GenotypeArray with the first n genotype in the shuffled GenotypeArray and the rest being shuffled non-EAS GenotypeArray. We calculated the Fst between this 2 shuffled GenotypeArray and note down the Fst (Fst-shuffled). FDR was applied to adjust for multiple testing. Subsequently, we applied additional filtering on the obtained FDR values to identify SVs with significant Fst. Specifically, we focused on events with an FDR threshold of less than 1% and an Fst value greater than the mean of the entire dataset.

We conducted a similar Fst analysis for 1000G-SV comparing EAS + SAS against other ancestry.

### Reporting summary
Further information on research design is available in the Nature Portfolio Reporting Summary linked to this article.

### Data availability
The CRAM files for the 34 1000 G samples used for benchmarking can be found in https://registry.opendata.aws/1000-genomes/. The VCF for the SVs called using long-read sequencing data for the 1000 G samples can be found in: https://ftp.1000genomes.ebi.ac.uk/vol1/ftp/data_collections/HGSVC2/release/v2.0/integrated_callset/variants_freeze4_sv_insdel_sym.vcf.gz. The VCF containing SV calls from gnomAD-SV can be retrieved from https://ftp.ncbi.nlm.nih.gov/pub/dbVar/data/Homo_sapiens/by_study/vcf/nstd166.GRCh38.variant_call.vcf.gz. The VCF containing SVs from 1000 G short-read data can be obtained from https://ftp.1000genomes.ebi.ac.uk/vol1/ftp/phase3/integrated_sv_map/supporting/GRCh38_positions/ALL.wgs.mergedSV.v8.20130502.svs.genotypes.GRCh38.vcf.gz. The sequence data used in this study were obtained under Data Access Application NPM00002 through the National Precision Medicine (NPM) Data Access Committee (DAC). The data are available under controlled access due to data privacy laws related to participant consent for data sharing. Bona fide researchers are required to submit a data access request outlining the proposed research, which will be subject to approval by the NPM DAC. The average processing time is 6-8 weeks. The data access request form and data access policy are available on the SG10K_Health web portal (https://npm.a-star.edu.sg/help/). The aggregated SG10K-SV-r1.4 VCF data can be downloaded via the CHORUS variant browser, which is accessible through registration with the SG10K_Health web portal (https://npm.a-star.edu.sg). The response time for access requests is approximately 3 working days, and the data will be available for download upon access approval. For more information, users can contact the NPM Programme Coordinating Office, A*STAR (contact_npco@gis.a-star.edu.sg).

### Code availability
Codes used for the analysis of the SG10K-SV dataset can be found in GitHub (https://github.com/c-BIG/SG10K-SV-MANUSCRIPT).

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

## Acknowledgements
This study made use of data generated as part of the Singapore National Precision Medicine program funded by the Industry Alignment Fund (Pre-Positioning) (IAF-PP: H17/01/a0/007 awarded to P.T.). This study made use of data / samples collected in the following cohorts in Singapore: The Health for Life in Singapore (HELIOS) study at the Lee Kong Chian School of Medicine, Nanyang Technological University, Singapore (supported by grants from a Strategic Initiative at Lee Kong Chian School of Medicine, the Singapore Ministry of Health (MOH) under its Singapore Translational Research Investigator Award (NMRC/STaR/0028/2017 awarded to J.C.C.) and the IAF-PP: H18/01/a0/016 awarded to J.C.C.); The Growing up in Singapore Towards Healthy Outcomes (GUSTO) study, which is jointly hosted by the National University Hospital (NUH), KK Women's and Children's Hospital (KKH), the National University of Singapore (NUS) and the Singapore Institute for Clinical Sciences (SICS), Agency for Science Technology and Research (A*STAR)(supported by the Singapore National Research Foundation under its Translational and Clinical Research (TCR) Flagship Programme and administered by the Singapore Ministry of Health's National Medical Research Council (NMRC), Singapore - NMRC/TCR/004-NUS/2008; NMRC/TCR/012-NUHS/2014. Additional funding is provided by SICS and IAF-PP H17/01/a0/005 awarded to N.K. and Y.S.C.); The Singapore Epidemiology of Eye Diseases (SEED) cohort at Singapore Eye Research Institute (SERI) (supported by NMRC/CIRG/1417/2015; NMRC/CIRG/1488/2018; NMRC/OFLCG/004/2018 awarded to C.Y.C.); The Multi-Ethnic Cohort (MEC) cohort (supported by NMRC grant 0838/2004; BMRC grant 03/1/27/18/216; 05/1/21/19/425; 11/1/21/19/678, Ministry of Health, Singapore, National University of Singapore and National University Health System, Singapore awarded to E.S.T.); The SingHealth Duke-NUS Institute of Precision Medicine (PRISM) cohort (supported by NMRC/CG/M006/2017_NHCS; NMRC/STaR/0011/2012, NMRC/STaR/0026/2015, Lee Foundation and Tanoto Foundation awarded to S.D. and K.K.Y.); The TTSH Personalised Medicine Normal Controls (TTSH) cohort funded (supported by NMRC/CG12AUG17 and CGAug16M012 awarded to K.P.L.). This research is also supported by the National Research Foundation (NRF) Singapore under its National Precision Medicine Program (NPM) Phase II Funding (MOH- 000588 awarded to P.T.) and administered by the Singapore MOH's National Medical Research Council (NMRC). The views expressed are those of the author(s) are not necessarily those of the National Precision Medicine investigators, or institutional partners. We thank all investigators, staff members and study participants who made the National Precision Medicine Project possible.

## Author contributions
T.H.J.J., L.Z.H., R.R., M.G.P., L.J.J., S.W.K.K., N.B., S.P. and P.T. contributed to the writing of the manuscript. T.H.J.J., L.Z.H., R.R., M.G.P., T.R.Y., N.B., R.T.J., S.X.L., T.Y.A. and L.W.K. contributed to the generation of figures and analysis of the data. T.H.J.J., L.Z.H., R.R. and M.G.P. contributed to the production and quality control of the SG10K-SV dataset. M.H., J.L.O., S.A., J.J., Y.S.C., T.H.L., L.L.G., Y.C.T., K.P.L., C.W.L.C., S.D., N.K., C.Y.C., J.C.C., E.S.T. and the SG10K_Health Consortium contribute to the production and collection of the data. All authors reviewed the manuscript.

## Competing interests
The authors declare no competing interests.

## Ethics
This project is approved by the NPM Data Access Committee with project ID: NPM00002.

## Additional information

[1]Genome Institute of Singapore, Agency for Science, Technology and Research, Singapore, Singapore. [2]Duke-NUS Medical School, Singapore, Singapore. [3]SingHealth Duke-NUS Institute of Precision Medicine, Singapore Health Services, Duke-NUS Medical School, Singapore, Singapore. [4]SingHealth Duke-NUS Genomic Medicine Centre, Duke-NUS Medical School, Singapore, Singapore. [5]Saw Swee Hock School of Public Health, National University of Singapore and National University Health System, Singapore, Singapore. [6]Department of Obstetrics & Gynaecology, Yong Loo Lin School of Medicine, National University of

Singapore, Singapore, Singapore. [7]Institute for Human Development and Potential (IHDP), Agency for Science, Technology and Research (A*STAR), Singapore, Singapore. [8]NHG Eye Institute, Tan Tock Seng Hospital, National Healthcare Group, Singapore, Singapore. [9]Personalised Medicine Service, Tan Tock Seng Hospital, Singapore, Singapore. [10]Singapore Eye Research Institute, Singapore National Eye Centre, Singapore, Singapore. [11]Centre for Innovation and Precision Eye Health, Yong Loo Lin School of Medicine, National University of Singapore, Singapore, Singapore. [12]Department of Cardiology, National Heart Centre Singapore, Singapore, Singapore. [13]Cardiovascular ACP, Duke-NUS Medical School, Singapore, Singapore. [14]SingHealth Duke-NUS Institute of Precision medicine, Singapore Health Services, Singapore, Singapore. [15]Cardiovascular and Metabolic Disorders Program, Duke-NUS Medical School, Singapore, Singapore. [16]Human Development, Singapore Institute for Clinical Sciences, Singapore, Singapore. [17]Clinical Data Engagement, Bioinformatics Institute, Agency for Science, Technology and Research, Singapore, Singapore. [18]Department of Biochemistry, Yong Loo Lin School of Medicine, National University of Singapore, Singapore, Singapore. [19]Population and Global Health, Nanyang Technological University, Lee Kong Chian School of Medicine, Singapore, Singapore. [20]Department of Epidemiology and Biostatistics, Imperial College London, London, UK. [21]Precision Health Research, Singapore, Singapore. [22]Department of Medicine, Yong Loo Lin School of Medicine, National University of Singapore, Singapore, Singapore. [23]Laboratory of Human Genomics, Genome Institute of Singapore (GIS), Agency for Science, Technology and Research (A*STAR), Singapore, Singapore. [24]Yong Loo Lin School of Medicine, National University of Singapore, Singapore, Singapore. [25]Hong Kong Genome Institute, Hong Kong, Hong Kong. [26]Department of Chemical Pathology, Chinese University of Hong Kong, Hong Kong, Hong Kong. [27]Lee Kong Chian School of Medicine, Nanyang Technological University, Singapore, Singapore. [28]Cancer Science Institute of Singapore, National University of Singapore, Singapore, Singapore. [46]Present address: Nalagenetics, Singapore, Singapore. [47]Present address: Human Genome Center, University of Tokyo, Bunkyō, Japan. [48]Present address: Translational Medicine, Sidra Medicine, Ar-Rayyan, Qatar. [49]These authors contributed equally: Joanna Hui Juan Tan, Zhihui Li, Mar Gonzalez Porta, Ramesh Rajaby.
✉e-mail: prabhakars@gis.a-star.edu.sg; tanbop@gis.a-star.edu.sg; Nicolas_Bertin@gis.a-star.edu.sg

## SG10K_Health Consortium

Khung Keong Yeo[29], Stuart Alexander Cook[29], Chee Jian Pua[29], Chengxi Yang[29], Tien Yin Wong[10], Charumathi Sabanayagam[10,30], Lavanya Raghavan[10], Tin Aung[10,30], Miao Ling Chee[10], Miao Li Chee[10], Hengtong Li[10,11], Jimmy Lee[31,32], Eng Sing Lee[33,34], Joanne Ngeow[19,35], Paul Eillot[36], Elio Riboli[36], Hong Kiat Ng[19], Theresia Mina[19], Darwin Tay[19], Nilanjana Sadhu[19], Pritesh Rajesh Jain[19], Dorrain Low[19], Xiaoyan Wang[19], Jin Fang Chai[5], Rob M. Van Dam[5,37,38], Yik Ying Teo[5], Chia Wei Lim[9], Pi Kuang Tsai[9], Wen Jie Chew[39], Wey Ching Sim[9], Li-xian Grace Toh[9], Johan Gunnar Eriksson[16,40], Peter D. Gluckman[41,42], Yung Seng Lee[16,43], Fabian Yap[44] & Kok Hian Tan[45]

[29]National Heart Research Institute Singapore, National Heart Centre Singapore, Singapore, Singapore. [30]Ophthalmology & Visual Sciences Academic Clinical Program, Duke-NUS Medical School, Singapore, Singapore. [31]Department of Psychosis, Institute of Mental Health, Singapore, Singapore. [32]Nanyang Technological University, Singapore, Singapore. [33]National Healthcare Group, Singapore, Singapore. [34]Nanyang Technological University, Lee Kong Chian School of Medicine, Singapore, Singapore. [35]National Cancer Centre, Singapore, Singapore. [36]School of Public Health, Imperial College London, London, UK. [37]Department of Nutrition, Harvard T.H. Chan School of Public Health, Boston, USA. [38]Exercise and Nutrition Sciences, Milken Institute School of Public Health, The George Washington University, Washington, USA. [39]Clinical Research & Innovation Office, Tan Tock Seng Hospital, Singapore, Singapore. [40]Department of Obstetrics & Gynaecology, Yong Loo Lin School of Medicine, NUS, Singapore, Singapore. [41]Singapore Institute for Clinical Sciences, Singapore, Singapore. [42]Liggins Institute, University of Auckland, Auckland, New Zealand. [43]Department of Paediatrics, National University Hospital, Singapore, Singapore. [44]Pediatrics, KK Women's and Children's Hospital, Singapore, Singapore. [45]Department of Obstetrics & Gynaecology, KK Women's and Children's Hospital, Singapore, Singapore.

