## [Peer Review File · Nature Communications]

A Catalogue of Structural Variation across Ancestrally Diverse Asian GenomesREVIEWER COMMENTS

Reviewer #1 (Remarks to the Author):

I have provided comments for author in a separated document.

Reviewer #2 (Remarks to the Author):

The manuscript entitled "A Catalogue of Structural Variation across Ancestrally Diverse Asian Genomes" presents a whole-genome sequencing (WGS) catalogue of structural variations (SVs) derived from thousands of Singaporeans of East Asia. This effort aims to address the under-representation of SVs reflective of Asian populations. The author claims that their study is one of the first and the most extensive multi-ancestry examination of SVs about Asian groups. The topic sounds very promising and impactful. However, I would like to point out that more citations on the current analysis of SVs can be included in the manuscript, including newly developed tools in characterizing SVs. The manuscript may also include a justification of why certain tools/software are used and why the bioinformatics pipelines are used to derive their results (comparative analysis with other methods and the choices of certain parameters/thresholds may be included). In addition, I have a few minor points that, if addressed, could further enhance the clarity and consistency of the manuscript:

1. In line 69, when ACMG is first mentioned, it would be helpful to provide its complete form, i.e., American College of Medical Genetics and Genomics (ACMG).
2. In line 89, the criterion for SVs should be denoted as " $\geq 50\text{bp}$ " instead of " $> 50\text{bp}$ ".
3. There appears to be some inconsistency regarding the spacing between numbers and units such as kb or Mb (as seen on line 229). Additionally, there's a mix of "kbp" and "kb". I recommend adopting a consistent notation throughout the paper.
4. In line 334, the first mention of 'Fst' would benefit from including its full name for clarity.
5. For all external data sources cited in the main text, it's crucial to specify the data's origin and offer detailed references in the data availability section. For instance, the reference to HGSC2 in line 701 needs this clarification.
6. It seems that Figure 5B and Figure 5C haven't been cited within the paper.
7. In line 456, please capitalize the 't' in "table 5" to maintain uniformity, making it "Table 5".
8. I observed that the content of the Methods and Materials section in the main manuscript mirrors that of the supplementary methods. Is this intentional?
9. The formatting, especially font size, and bolding, within the supplementary figure legends lacks consistency.
10. In line 815, I'm curious about the choice of the 1Mb window. Would employing a smaller 100kb window have significantly altered the results? Was this decision influenced by the size distribution of the detected SVs?

Reviewer #1 (Remarks to the Author):

General comment:

This manuscript reports SVs that are discovered from multi-ancestry Singapore individuals and describes their functional and clinical significance by integrating publicly available resources and information. While the authors performed multiple analyses with large-scale data (8,392 Singaporeans of East Asian, Southeast Asian, and South Asian ancestries, which is a novelty of this manuscript), they failed to draw significant biological meanings from their analyses efficiently. To publish this manuscript as an original research article in Nature Communications rather than an article to describe resources, this point should be addressed in the revised manuscript through in-depth and more focused analyses.

Our response: We are grateful for the valuable suggestions and feedback from the reviewers which we believe have improved our manuscript and our SG10K-SV catalogue. We hope the clarifications and manuscript's revision described below will address all the comments raised.

Major comment #1:

The authors divided the whole dataset into three parts, but I could not find a clear reason why they did it in the main text, what criteria was used to divide the dataset and what is the logic behind this selection. It is a crucial point as the diversity of samples and their sequencing depth are directly related to the discovery of SVs. Please make this point clear in the main text.

Our response: We agree with the reviewer that the initial manuscript did not motivate clearly why we elected to split the entire collection of WGS libraries into three parts (a "discovery cohort", a "30x validation" and a "15x validation" cohorts). We have edited the text in page 6 line 155 - 167 to make this point clear.

"The SG10K-SV-r1.4 dataset comprises multiple sub-cohorts sequenced at heterogeneous depths and using different library construction methods (Supplementary Table 1). Previous studies have demonstrated that library preparation methods, PCR-free (PCR-) and PCR-amplified (PCR+), can cause non-uniformity of sequencing coverage¹⁰, which can in turn affect the ability to accurately detect structural variation. Differences in sequencing depth between libraries within a collection also impact structural variation genotyping sensitivity. To ensure robust SV analysis and to reduce technical confounding factors, we split the collection into three datasets, namely (1) Discovery cohort of 5,487 individuals (average sequencing depth: 15x, library construction method: PCR+), (2) 15x_validation cohort containing 1,523 individuals (average sequencing depth: 15x, library construction method: PCR-), (3) 30x_validation cohort consist of 1,922 individuals (average sequencing depth: 30x, library construction method: PCR+)."

Major comment #2:

As the authors mentioned in the text, 15x coverage might not be enough to discover SVs at high sensitivity, and indeed, the metrics in supplementary table 6 do not look good (precision as well). Unless you can increase coverage to complement this issue,

I would suggest clearly stating what percentage of SV would be missing and incorrectly called in the main text with support from a larger set of benchmark analyses than the authors already did for readers.

Our response: We agree with the reviewer that it would be of interest to the readership to be provided with a benchmark-backed analysis of the percentage of that would be missing or incorrectly called because of 15x coverage.

We leveraged on the collection of 34 long-read sequenced 1000genomes libraries as a truth set to estimate the fraction of missing (False Negative) or incorrectly called (False Positive) SV calls from matching 30x and 15x (down-sampled) short reads sequencing data using our SV discovery pipeline.

We found an average 25.3% decrease in the number of False Positive (that is SV identified using long-reads that were also discovered using short-reads for a given libraries), between 30x and 15x down-sampled short-reads-derived libraries. We also found an average 14.6% increase in the number of False Negative (that is SV identified using long-reads that were not discovered using short-reads for a given libraries), between 30x and 15x down-sampled short-reads-derived libraries. We included the number of variants that will be missed by the pipeline in the main text page 7 line 191-196.

“This benchmarking also allowed us to estimate the fraction of SVs missed by our SV detection pipeline between 15x and 30x WGS. On average, across all the 1000 genomes samples, 14.6% of long-read-defined SVs re-identified when sequenced at a depth of 30x could not be re-identified when down-sampled to 15x (Supplementary Fig. 2).”

Supplementary Fig. 2 True positive, false positive and false negative counts for Manta, Delly, Smoove and their combination for all classed of SVs. **a** Boxplot showing the number of false positive counts between 15x and 30x coverage for each SV caller. Combined refers to variants that are detected in all four pipelines. **b** Boxplot showing the false negatives counts between 15x and 30x coverage for each SV caller. **c** Boxplot showing the true positive counts between 15x and 30x coverage for each SV caller.

Major comment #3:

While the authors are trying to consider the different characteristics across SV types by using different algorithms to discover different types of SVs (Manta for insertion/deletion, MELT for mobile element insertion, and SurVindel2 for duplication), discovering SVs using only one algorithm for each SV type may not be enough in terms of false positives. As each algorithm has different biases depending on the underlying sequence context of the variants and the data properties, support from multiple algorithms would enhance the accuracy of SV discovery. Therefore, I suggest using at least three different algorithms for each SV type (at least for insertion and deletion).

Our response: While as noted by the reviewer, care was taken to integrate several SV callers, each better suited to insertions, deletions, or duplications specific characteristics, considering the integration of multiple independent insertion and deletion SV callers to improve the accuracy of their discovery is an interesting suggestion.

We leveraged on the collection of 34 long-read sequenced 1000genomes libraries as a truth set to estimate insertion and deletion SV calls' precisions, recalls and F1-scores derived from analyzing long-read matching 30x and 15x (down-sampled) short reads

sequencing data using three alternative callers (in addition to manta used for insertion and deletion SV discovery in our SG10K-SV pipeline) : Delly, Smoove, as well as combining the calls of all three callers (Manta, Delly and Smoove) using SVimmer-Graphtyper2.

Details of this benchmarking of multiple algorithms have been added as Supplementary Note 1 and mentioned in the main text page 7 line 184-191.

“we benchmarked several well-known SV callers, including Manta²⁶, Delly²⁷ and Smoove²⁸. SVs identified using long-read WGS in thirty-four 1000 genome samples by Ebert et al.²⁹ were used as a truth set to assert the performance of each SV caller to recover joint-genotyped SVs across matched 30x and 15x down-sampled short-read WGS (Supplementary Note 1, Supplementary Table 2). While measures of precision for Delly were superior to that obtained with Manta, Manta yielded overall higher F1-scores than other tools individually or in combination (Fig. 1c, d and e and Supplementary Fig. 2).”

Fig. 1: SG10K-SV-r1.4 Structural Variant landscape. a Number of Asian samples in SG10K-SV-r1.4 compared to (short-read derived) 1000 genomes SV, gnomAD-SV and CCDG reference studies. **b** SG10K-SV-r1.4 analysis pipeline diagram. **c** Benchmarking of various SV tools for SV detection. Boxplot showing the precision between 15x and 30x coverage for each SV caller. Combined refers to variants that are detected in all four pipelines. **d** Boxplot showing the recall between 15x and 30x coverage for each SV caller. **e** Boxplot showing the F1-score between 15x and 30x coverage for each SV caller.

In short, while individually some of the algorithms provided better precision than manta for deletion rediscovery, recall rates were inferior, yielding less favourable F1-Scores. Combining all three algorithms did not provide substantial improvement compared to using manta alone for SV rediscovery.

The results of this benchmarking provide, we believe, the readership with a reassurance that our SG10K-SV discovery pipeline which albeit only leveraging upon manta to identify insertions and deletions (duplication and mobile element insertion, which manta is not ideally suited for, are complemented by SurvIndel2 and MELT respectively) is sufficiently accurate and does not introduce any excess of False Positive SVs compared to an approach in which multiple independent insertion and deletion SV callers would be combined.

Minor comment #1:

Most part of the first section of the Result titled “SV Catalogues of Three Major Ancestry Groups” describes methodological details. This should be moved to the Methods section.

Our response: We agree with the reviewer and have moved most of the section into the Methods section. For this paragraph, we now focused more on describing the dataset and the benchmarking of various SV algorithms.

Minor comment #2:

Line 204-205 It needs more details about the SVs discovered in the validation dataset. e.g., How many and what types of SVs are detected in these datasets?

Our response: We have included in the text the number and the types of SVs detected in the validation datasets in page 7-8 line 210-216:

“Utilizing variants in the discovery dataset, we genotyped these variants in samples from our two validation datasets to ensure that results observed in the discovery dataset are reproducible. 40,883 and 60,715 of the SVs detected in the discovery dataset were identified in the 15x PCR- and 30x PCR+ validation dataset, respectively. For the 15x_validation dataset, a total of 6,775 deletions, 17,036 duplications, and 17,072 insertions were detected. In the 30x_validation dataset, 19,275 deletions, 21,377 duplications and 20,063 insertions were detected.”

Minor comment #3:

Line 261-274 This paragraph includes many overinterpretations, which should be toned down and moved to the Discussion section.

Our response: We agree with the reviewer, and we have tone down the paragraph in page 9 -10 line 271-281:

“Common deletions ($AF \geq 1\%$) were significantly depleted at putative enhancers and insulators, consistent with a model of negative selection acting on alterations affecting gene expression (Fig. 3a). In contrast, rare ($1\% > AF \geq 0.1\%$) and ultra-rare ($AF < 0.1\%$) deletions did not exhibit similar depletion signals. Common duplications were also significantly depleted at distal and proximal enhancers (Fig. 3a) again suggesting the action of purifying selection. Unexpectedly, we observed common duplications being enriched at annotated non-promoter H3K4me3 regions. To deepen this observation, we examined the intersect of 81 non-promoter H3K4me3 regions overlapping common duplications, and found that they were highly and significantly enriched for tandem repeats relative to all 25,537 H3K4me3 regions (fold enrichment: 4.6 : hypergeometric p-value: 2.45×10^{-23}). ”

Minor comment #4:

Line 281-285 It does not necessarily mean some level of error in ChIP-seq peaks. As short reads were used similarly, the SV discovery might fail to call SV correctly in these regions. It needs more evidence to support this idea.

Our response: We agree with the reviewer, and we have tone down this part in page 9-10 line 281 - 283:

“We speculate that since read mapping artifacts are common at tandem repeats, it is possible that these mapping artifacts might have contributed to artefactual ChIP-seq peaks at these tandem repeat regions.”

**Minor comment #5:
Line 325 Figure 2A -> Figure 1E**

Our response: We thank the reviewer for identifying the incorrect figure was referenced in the main text and have amended the manuscript accordingly (please note that together with other modifications to the text and figures suggested during the revision process, the figure is now numbered 2a (page 11 line 323-325):

“Reflecting the novelty of the SG10K-SV catalogue, 66.5% (49,601/73,035) and 86.7% (63,367/73,035) of the SVs identified were not previously reported in gnomAD-SV (Fig. 2a) or 1000G-SV catalogues (Fig. 2b), respectively.”

**Minor comment #6:
Line 332 Why is the word “event” added to SV from this part onwards? Is there any specific reason?**

Our response: We thank the reviewer for identifying this shortcoming. To be less repetitive, we have, when unambiguous, referred to SVs as “events”. The mentions of “SV event(s)” were an oversight and have been corrected in the text, and are now referring to Structural Variation(s) as either, simply, “SV(s)” or “event(s)”

**Minor comment #7:
Line 332-337 It might not be a good idea to use the gnomAD-SV dataset for discovering Asian-specific SVs because the gnomAD dataset is not diverse as much as other large-scale datasets (e.g., 1000 genomes project and human genome diversity project).**

Our response: We thank the reviewer for this insightful comment and agree that the relative lack of ancestral diversities within the gnomAD-SV catalogue compared to those derived from smaller in size but much more diverse, such as 1000 genomes or human genome diversity projects, makes it a suboptimal dataset against which SV’s Asian specificity could be asserted.

We have amended this section of the text, complementing the gnomAD-SV catalogue based ancestry prevalence analysis with a similar analysis of ancestry prevalence in light of the 1000 Genomes Project phase 3 SV catalogue in page 7 line 205-207.

“Approximately 66.5% and 86.7% of SG10K-SV-r1.4 events were novel (Fig. 2a, b) with respect to gnomAD-SV¹⁰ and 1000 Genomes Project phase 3 SV¹² (1000G-SV), respectively,”

And page 11 line 323-331

“Reflecting the novelty of the SG10K-SV catalogue, 66.5% (49,601/73,035) and 86.7% (63,367/73,035) of the SVs identified were not previously reported in gnomAD-SV (Fig. 2a) or 1000G-SV catalogues (Fig. 2b), respectively. In total, 47,770 SVs in SG10K-SV did not overlap with either study (1000G-SV and gnomAD-SV). Applying a call rate cut-off across each ethnic group of $\geq 50\%$ within SG10K-SV, we identified 42,239 SVs and hereby termed these as “Asian specific - novel” SVs. The majority of novel Asian-specific SVs identified exclusively in our catalogue exhibited lower allele frequencies than SVs identified in both SG10K-SV and gnomAD-SV or SG10K-SV and 1000G-SV (Supplementary Fig. 8).”

Minor comment #8:

Line 359-362 What does the clustering look like if you use SNVs here?

Our response: We thank the reviewer for this thoughtful suggestion and are now providing in supplementary material (Supplementary Figure 9) a scatterplot of the two main PCA components obtained from single nucleotide polymorphisms (SNVs) identified in the very same samples to those included in SG10K-SV.

Supplementary Fig. 9 Scatter plot of the top-2 principal components of a SG10K_Health dataset Single Nucleotide Variant based PCA analysis showing the population structure in the Singaporean population.

The manuscript was amended, highlighting to the readership the similarity of the SV and SNV respective principal component analysis and the ability of the SV-derived PCA analysis to uncover the underlying ancestry-specific population clustering and structure. The text can be found in page 12 line 349-350.

“similar to SNV clustering using the SG10K_Health²³ dataset with the same samples (Supplementary Fig. 9).”

Minor comment #9:

Line 388-392 It sounds off-topic and does not need to be mentioned in the text.

Our response: We agree with the reviewer and have removed these statements from this section of the manuscript.

Minor comment #10:

Line 448 Why did you use 1Mbp? Please describe it in the Methods.

Our response: We believe using a window size of 1Mb provides a good balance between being large enough to capture most LD between SVs and SNPs without being computationally overwhelming. Notably this 1Mb window size choice was also made by the TOPMed consortium (Jun G et al Res Sq [Preprint] 2023) upon similar computation/analysis of SNPs and SVs linkage disequilibrium.

We have amended the text (page 27 line 807-809) accordingly as follows:

“ We compute LD between high-confidence (call rate ≥ 0.8) common ($MAF \geq 1\%$) SVs ($n=6,772$) and small variants ($n=9,450,184$) located within a 1Mb window using PLINK v1.9⁸¹, similar to the approach used by TopMed⁷⁵). ”

Minor comment #11:

Line 540 ...indicating that this duplication (is) common...

Our response: This example has been removed from the text after the re-generation of the release 1.4.

Minor comment #12:

Line 586 Please describe the full word of MAD when it first appears in the text.

Our response: We included the full word on MAD in the main text page 19 line 546-547:

“It also included QC checks intended to discard samples with poor sequencing quality (e.g. hard filters for error rate and contamination), unusual numbers of calls (e.g. Median absolute deviation (MAD)-based filters on het/hom ratio), chromosome aneuploidies, and/or samples with related individuals in the same cohort (see methods in Wong et al., 2023 for additional details).”

Minor comment #13:

Line 597 duplicates ->PCR duplicates?

Our response: We edited the Main text to include PCR duplicates in page 19 line 557:

“median autosome coverage”: The median coverage in autosomes, excluding (i) bases in reads with low mapping quality (mapq < 20); (ii) bases in reads marked as PCR duplicates, and (iii) overlapping bases in read pairs; calculated with mosdepth⁶⁹.”

Minor comment #14:

Line 605 What is “PF reads” here?

Our response: We included the definition of PF reads in page 19 line 566-567:

“pct reads aligned”: The percentage of PF reads that align to the reference; calculated with picard AlignmentSummaryMetric⁷⁰. PF reads refer to reads that passes Illumina’s filter.”

Minor comment #15:

Line 607-608 Please describe the exact definition of proper pairs here.

Our response: We described the exact definition of proper pairs in the text page 19-20 line 568-571:

“pct reads properly paired”: The percentage of reads that align as proper pairs as calculated with samtools stats⁷¹. Properly pair reads are reads in which both reads in the pair are mapped and they are mapped within the range from each other based on the estimated insert size distribution”

Minor comment #16:

Line 685 How did you merge duplications here? Did you merge duplications that have the exact same position and size? It may be critical because the resulting SV set really depends on how to merge them in some cases.

Our response: Akin the merging of insertion or deletion SVs detected across samples using SVimmer and in line with the recommendation of SurVIndel2 authors, duplication events were considered identical events and merged whenever their extremities were located within 200bp and their length difference were within 100bp. The main manuscript text (Page 22 line 653-656) was amended to provide this information to the SG10K-SV manuscript readership.

“Then, we clustered the duplications as recommended in the manuscript of SurVIndel2, merging events whose length differ by less than 100bp, and whose extremities were located within 200bp of each other in a manner analogous to that employed by SVimmer for insertion and deletion clustering.”

Minor comment #17:

Line 694 How about the other regions that are hard to map? (e.g., complex regions) Especially for short reads, the mappability should be considered in the filtering step of SVs to reduce false positives.

Our response: We agree with the reviewer that it would be of interest to the readership to be provided with a more detailed account of the region of the genome where short read sequences mapping is typically problematic and were, as is common practice, excluded from our analysis. The method section of the manuscript (Page 23 line 678 - 681) was amended accordingly as follows:

“(i) retain events in autosomal contigs (chr1-22), (ii) exclude those that occur in centromeres, telomeres, heterochromatin region²⁷, (iii) exclude regions in the primary assembly that overlap with ALT contigs and (iv) exclude N-masked regions of the reference genome.”

Minor comment #18:

Line 702 The paper includes CLR reads as well, which were used for most of the assemblies.

Our response: We thank the reviewer for bringing this information to our attention. Indeed, in Ebert *et al* (Ebert, P. *et al*. Science 2021) assemblies are derived from both HiFi and CLR long-reads approaches. We have amended the manuscript text (Page 23 line 685-687) to accurately describe the use of both HiFi and CLR long-reads by Ebert *et al*.

“*The Human Genome SV Consortium (HGSC) released HGSC2, a comprehensive set of SVs detected in 35 samples in the 1000 Genome Project using PacBio HiFi and CLR reads²⁹.*”

Minor comment #19:

Line 729 What does the Hard-Weinberg Equilibrium (HWE) normalization mean exactly here?

Our response: We agree with the reviewer that this description of the approach we used to compute principal components using `hail hwe_normalized_pca` function on the SV-derived genetic relationship matrix was ambiguous.

For reference, the detailed description of this “`hl.hwe_normalized_pca()`” function can be found in:

https://hail.is/docs/0.2/methods/genetics.html#hail.methods.hwe_normalized_pca

The text (Page 23 line 703-704) has been modified to alleviate any ambiguity:

“*To investigate the relationship between the different ethnic groups in Singapore, we performed principal component analysis (PCA) using all variants (deletions, insertions, duplications and MEIs) genotypes using the “hl.hwe_normalized_pca()” function in Hail⁷⁵.*”

Minor comment #20:

Line 831-833 Please describe the details of permutation analysis here. (e.g., the number of replicates)

Our response: As per the reviewer request, the relevant method section (page 27 line 827-832) was amended to describe in greater details how the permutation based Fst statistical significance analysis was performed:

“This approach involved maintaining the original genotype matrix while randomly shuffling the ancestry labels across 1000 iterations, for each of which the Fst was recalculated. The significance of the observed Fst values was then determined by comparing these against the distribution of Fst values obtained from the permuted data, calculating a p-value based on the proportion of permuted Fst values lower than the observed value. FDR was applied to adjust for multiple testing.”

Reviewer #2 (Remarks to the Author):

The manuscript entitled “A Catalogue of Structural Variation across Ancestrally Diverse Asian Genomes” presents a whole-genome sequencing (WGS) catalogue of structural variations (SVs) derived from thousands of Singaporeans of East Asia. This effort aims to address the under-representation of SVs reflective of Asian populations. The author claims that their study is one of the first and the most extensive multi-ancestry examination of SVs about Asian groups. The topic sounds very promising and impactful. However, I would like to point out that more citations on the current analysis of SVs can be included in the manuscript, including newly developed tools in characterizing SVs. The manuscript may also include a justification of why certain tools/software are used and why the bioinformatics pipelines are used to derive their results (comparative analysis with other methods and the choices of certain parameters/thresholds may be included). In addition, I have a few minor points that, if addressed, could further enhance the clarity and consistency of the manuscript:

Our response: We thank the reviewer for his/her insightful observations, and we appreciate the constructive and insightful comments of the reviewers for providing valuable inputs to enhance the clarity and consistency of the manuscript.

We agree with the reviewer that it would be of interest to the readership to include a justification of the choice of tools/software/pipeline used and to provide a comparative analysis with other methods. We have added a Supplementary note section in which the approach and choices leading to our SG10K-SV analysis pipeline design are compared to potential alternative approaches.

We leveraged on the collection of 34 long-read sequenced 1000genomes libraries as a truth set to estimate insertion and deletion SV calls' precisions, recalls and F1-scores derived from analyzing long-read matching 30x and 15x (down-sampled) short reads sequencing data using two manta (the algorithm used for insertion and deletion SV discovery in our SG10K-SV pipeline) alternative callers : Delly, Smoove, as well as combining the calls of all three callers (Manta, Delly and Smoove tools) using SVimmer-Graphyper2.

Details of this benchmarking of multiple algorithms for each insertion and deletion SV type have been added as supplementary note and mentioned in the main text page 7 line 184-196.

“we benchmarked several well-known SV callers, including Manta²⁶, Delly²⁷ and Smoove²⁸. SVs identified using long-read WGS in thirty-four 1000 genome samples by Ebert et al.²⁹ were used as a truth set to assert the performance of each SV caller to recover joint-genotyped SVs across matched 30x and 15x down-sampled short-read WGS (Supplementary Note 1, Supplementary Table 2). While measures of precision for Delly were superior to that obtained with Manta, Manta yielded overall higher F1-scores than other tools individually or in combination (Fig. 1c, d and e and Supplementary Fig. 2). This benchmarking also allowed us to estimate the fraction of SVs missed by our SV detection pipeline between 15x and 30x WGS. On average, across all the 1000 genomes samples, 14.6% of long-read-defined SVs re-identified

when sequenced at a depth of 30x could not be re-identified when down-sampled to 15x (Supplementary Fig. 2)."

In short, while individually some of the algorithms provided better precision than manta for deletion rediscovery, recall rate were inferior, yielding a less favourable F1-Score. Combining all three algorithms did not provide substantial improvement to using manta alone for insertion or deletion re-discovery.

The result of this benchmarking provides, we believe, the readership with a reassurance that our SG10K-SV discovery pipeline which albeit only leveraging upon manta to identify insertions and deletion (duplication and mobile element insertion, which manta is not ideally suited for are complemented by SurvIndel2 and MELT respectively) is sufficiently accurate and does not introduce any excess of False Positive SVs compared to an approach in which multiple independent insertion and deletion SV callers would be combined.

Below we address each of the specific minor points and comments brought to our attention by the reviewer.

1. In line 69, when ACMG is first mentioned, it would be helpful to provide its complete form, i.e., American College of Medical Genetics and Genomics (ACMG).

Our response: We have amended the abstract and provided the first reference to the American College of Medical Genetics and Genomics (ACMG) in complete form in page 10 line 307 – 310.

"We assessed the potential impact of SVs on major clinically actionable genes, focusing on 81 American College of Medical Genetics and Genomics (ACMG v3.2⁴⁴) defined actionable genes associated with highly penetrant and actionable genetic conditions"

2. In line 89, the criterion for SVs should be denoted as " ≥ 50 bp" instead of " > 50 bp".

Our response: We agree and have amended the text (Page 4 line 91-94) accordingly:

"SVs are genome rearrangements ≥ 50 bp and can be classified into different classes such as deletions, duplications, insertions (including mobile element insertions), translocations and inversions¹⁰."

3. There appears to be some inconsistency regarding the spacing between numbers and units such as kb or Mb (as seen on line 229). Additionally, there's a mix of "kbp" and "kb". I recommend adopting a consistent notation throughout the paper.

Our response: For consistency, we edited the spacing between

“While most detected SVs were small (Fig. 2c), we identified 2,678 deletions and 2,065 duplications longer than 10kb. There was a striking abundance of SVs at 300bp, 2kb and 6kb (Fig. 2c). The 300bp and 6kb insertions corresponded to Alu and LINE1 elements respectively, the two most abundant classes of transposable elements in the human genome (~11%³⁴ and ~17%³⁵ of the genome). The 2kb SVs represent composite SVA (SINE, Variable Number Tandem Repeat, and Alu) transposons.”

We also edited the notation to keep it consistent throughout the paper.

“Together, these 251 regions affected ~211Mb, in line with previous findings²⁹. Notably, 36% (90 out of 251) of the hotspot regions were located within 5Mb of the ends of the chromosomes as well as near the centromeric regions”

“Copy number gain events were typically larger compared to LOF events (median size 90kb vs 9.7kb).”

“For example, we identified a 9.4kb deletions in three Chinese individuals (AF = 5.18×10^{-4}), affecting TRDN (Fig. 3d)”

“We also found a 9.16kb heterozygous deletion affecting PRKAG2 present in two Chinese individuals”

“Notable examples of Asian-specific SVs include a previously reported 2.9kb deletion (chr2:111125617-111128520) in intron 2 of the BIM gene”

“Another example comprises a rare 19.3kb deletion (chr16:165396_184700) spanning the HBA1 and HBA2 genes”

“we observed a 2.7kb deletion (chr6:8432262-8434992) in SLC35B3 gene”

“Another example was a rare 8.8kb deletion (chr6:158745097-158753965) overlapping STYL3”

“A third example was a 9.8kb duplication overlapping PROCR, which encodes a receptor for activated protein C⁵³.”

“We also identified a 18kb duplication (chr6:73747426-73766255) overlapping CD109, a glycosylphosphatidylinositol (GPI) anchored protein⁵⁴”

“we identified a 6.9kb deletion overlapping TRIM48”

“LD was computed for high-confidence (call rate ≥ 0.8) common ($MAF \geq 1\%$) SVs ($n=6,772$) and small variants ($n=9,450,184$) located within a 1Mb distance (Fig. 5a).”

4. In line 334, the first mention of ‘Fst’ would benefit from including its full name for clarity.

Our response: We included the full name of Fst (page 12 line 361-363) at the first mention of it:

“To gain a more granular understanding of ancestry-specific SV patterns, we calculated fixation indexes (F_{st})⁵⁰ for each of the detected SV and assigned a significance score to each observation using permutation analysis (see Methods).”

5. For all external data sources cited in the main text, it's crucial to specify the data's origin and offer detailed references in the data availability section. For instance, the reference to HGSVC2 in line 701 needs this clarification.

Our response: We thank the reviewer for bringing this omission to our attention. We have included the source of the data in the data availability section (page 29, line 881-896).

“Data Availability

The CRAM files for the 34 1000G samples used for benchmarking can be found in <https://registry.opendata.aws/1000-genomes/>.

The VCF for the SVs called using long read sequencing data for the 1000G samples can be found in:

https://ftp.1000genomes.ebi.ac.uk/vol1/ftp/data_collections/HGSVC2/release/v2.0/integrated_callset/variants_freeze4_sv_insdelsym.vcf.gz.

The VCF containing SV calls from gnomAD-SV can be retrieved from https://ftp.ncbi.nlm.nih.gov/pub/dbVar/data/Homo_sapiens/by_study/vcf/nstd166.GRCh38.variant_call.vcf.gz.

The VCF containing SVs from 1000G short-read data can be obtained from https://ftp.1000genomes.ebi.ac.uk/vol1/ftp/phase3/integrated_sv_map/supporting/GRCh38_positions/ALL.wqs.mergedSV.v8.20130502.svs.genotypes.GRCh38.vcf.gz.”

6. It seems that Figure 5B and Figure 5C haven't been cited within the paper.

Our response: We thank the reviewer for bringing this omission to our attention and are now including reference to Figure 5b (page 14-15 line 447-452) and Figure 5c (page 15 line 454-457) in the manuscript's main text.

“This SV was in strong LD with two GWAS lead SNPs (rs4085613 ($R^2=0.97$) and rs4845459 ($R^2=0.98$); Fig. 5b)^{63,64}. Notably, both SNPs are associated with psoriasis ($P=7 \times 10^{-30}$ and $P=6 \times 10^{-11}$) in individuals of East Asian ancestries. Both SNPs are not found in the coding region of the genes and hence, our analysis suggests that the linked LOF SV should also be considered a potential causal variant for psoriasis in this locus”

“We also observed a predicted LOF SV (chr11:55,264,123-55,271,064) deleting exons 2 to 6 of TRIM48 exhibited strong LD ($R^2=0.903$; Fig. 5c) with an intergenic GWAS-lead SNP (chr11:54,697,371; rs11532186) associated with altered glomerular filtration rate.”

7. In line 456, please capitalize the 't' in "table 5" to maintain uniformity, making it "Table 5".

Our response: We edited the manuscript and not the table has been changed to Table 7 (page 14 line 438) and capitalize the t:

“Supplementary Table 7 lists all 385 SG10K-SVs candidate causative genetic alteration together with their associated GWAS lead SNPs.”

8. I observed that the content of the Methods and Materials section in the main manuscript mirrors that of the supplementary methods. Is this intentional?

Our response: We removed the content in the supplementary that mirrors the methods and materials section. We initially thought that the methods section will be under the supplementary methods.

9. The formatting, especially font size, and bolding, within the supplementary figure legends lacks consistency.

Our response: We thoroughly reviewed references made to figures, tables and have aligned with Nature Communication elements' formatting (bold, uppercasing of the first letter and lower casing of figure panel's references, ...).

10. In line 815, I'm curious about the choice of the 1Mb window. Would employing a smaller 100kb window have significantly altered the results? Was this decision influenced by the size distribution of the detected SVs?

Our response: We believe using a window size of 1Mb provides a good balance between being large enough to capture most LD between SVs and SNPs without being computationally overwhelming. Notably this 1Mb window size choice was also made by the TopMed consortium (Jun G *et al*/ Res Sq [Preprint] 2023) upon similar computation/analysis of SNPs and SVs linkage disequilibrium.

The text (Page 27 line 807-809) was amended as follows

“We compute LD between high-confidence (call rate ≥ 0.8) common (MAF $\geq 1\%$) SVs (n=6,772) and small variants (n=9,450,184) located within a 1Mb window using PLINK v1.9⁸¹, similar to the approach used by TopMed⁷⁵). ”

Reviewer #1 (Remarks to the Author):

The revised manuscript is greatly improved in terms of clarity and presentation, and especially the benchmarking analyses are logical. I still remain a bit skeptical about the SV/gene candidates they listed up in the paper, but otherwise feel that this paper presents a valuable resource that includes underrepresented populations to the research community. I have a minor comment about the LD analysis: As the authors stated, I believe that the window size of 1Mb is large enough, but I don't think it is a good justification that some other paper has used the same value.

Our response: We thank the Reviewer for the precious time in reviewing our manuscript and providing valuable feedback, allowing us to improve the quality of our manuscript.

Regarding our choice of a 1Mb LD lookup window and its justification (i.e. aligning with a similar choice made by the TOPMed consortium -- Jun G *et al.* Structural variation across 138,134 samples in the TOPMed consortium. Res Sq 2023), we agree that while referencing a previously used method can be helpful for context, it does not in itself provide a strong justification. However, we note that only 14 out of the 894 LD associations with an $R^2 > 0.8$ were more than 500 kb apart. Thus, while using a larger window might have revealed additional GWAS lead-SNP and SV associations, the 1Mb window likely captured most of the relevant GWAS lead-SNP SG10K-SV associations effectively.

Reviewer #2 (Remarks to the Author):

The revision of the manuscript “A Catalogue of Structural Variation across Ancestrally Diverse Asian Genomes” has greatly improved the clarity and overall quality of the work.

Our response: We thank the Reviewer for the precious time in reviewing our manuscript and providing valuable feedback, allowing us to improve the quality of our manuscript.

Other than these minor points raised below, the revision has addressed all the other comments.

1. Page 6 line 160, change “structural variation” to “the structural variation”;

Our response: We have amended the main text in page 6 line 164-165:

“Previous studies have demonstrated that library preparation methods, PCR-free (PCR-) and PCR-amplified (PCR+), can cause non-uniformity of sequencing coverage¹⁰, which can in turn affect the ability to accurately detect the structural variation.”

2. Page 7 line 186, the number “thirty-four” should be revised to “34” for consistency;

Our response: We have amended the main text in page 7 line 191:

“SVs identified using long-read WGS in 34 1000 genome samples by Ebert et al.²⁹ were used as a truth set to assert the performance of each SV caller to recover joint-genotyped SVs across matched 30x and 15x down-sampled short-read WGS (Supplementary Note 1, Supplementary Table 2).”

3. In the legend of Supplementary Fig. 2 line 5, the term “false negatives” should be corrected to “false negative”;

Our response: As we have added the benchmarking for each structural variation type, Supplementary Fig. 2 has become Supplementary Fig. 3. The text has been edited to:

“Supplementary Fig. 3 True positive, false positive and false negative counts for Manta, Delly, Smoove and their combination for all classed of SVs using 34 1000G samples with two different sequencing depth (15x and 30x coverage).
a Boxplot showing the number of false positive counts between 15x and 30x coverage for each SV caller. Combined refers to variants that are detected in all four pipelines. b Boxplot showing the false negative counts between 15x and 30x coverage for each SV caller. c Boxplot showing the true positive counts between 15x and 30x coverage for each SV caller. The boxplots showed in a-c display the median and first/third quartiles.”

4. The previous review requested a benchmarking of at least three different algorithms for each structural variation (SV) type. However, the revision provided does not include detailed comparison results for each SV type individually. It is important to present these results separately rather than collectively to better assess the performance of each algorithm across different SV types;

Our response: We have now included detailed comparison results for each individual SV type in Supplementary Fig. 2, as well as a description of the results in Supplementary Note 1. Similar to the results we obtained when looking at all SVs collectively, we noticed that Delly+Graphtyper2 has a higher precision than other tools for deletions and insertions. However, Manta+Graphtyper2 has a higher recall and F1-score than other tools.

Supplementary Fig. 2 Benchmarking of various SV callers for deletions and insertions using 34 100G samples with two different sequencing depths.
a Boxplot showing the precision for deletions between 15x and 30x coverage for each caller. Combined refers to variants that are detected in all four pipelines. **b** Boxplot showing the recall for deletions between 15x and 30x coverage for each caller. **c** Boxplot showing the F1-score for deletions between 15x and 30x coverage for each caller. **d** Boxplot showing the precision for insertions between 15x and 30x coverage for each caller. **e** Boxplot showing the recall for insertions between 15x and 30x coverage for each caller. **f** Boxplot showing the F1-score for insertions between 15x and 30x coverage for each caller. The boxplots in a-f display the median and first/third quartiles.

5. Regarding the rebuttal for minor comment #16, is there any reason to select 100 bp and 200 bp thresholds to merge the duplications?

Our response: The thresholds were chosen to allow for a degree of flexibility as duplications tend to occur in tandem repetitive regions, which can cause technical challenges for detection, such as, shifting breakpoints when detecting duplications using split reads in these noisy regions. For detailed information on the tool used for duplication detection, please refer to this manuscript: <https://www.biorxiv.org/content/10.1101/2023.04.23.538018v2.full>